health and disease and epidemiology

dengue, Zika, chikungunya, cluster analysis, spatio-temporal analysis

**Author for correspondence:**
Laís Picinini Freitas
e-mail: lais.picinini.freitas@gmail.com

# Space–time dynamics of a triple epidemic: dengue, chikungunya and Zika clusters in the city of Rio de Janeiro

Laís Picinini Freitas[1], Oswaldo Gonçalves Cruz[2], Rachel Lowe[3,4,5] and Marilia Sá Carvalho[2]

[1]Escola Nacional de Saúde Pública Sergio Arouca (ENSP), and [2]Programa de Computação Científica (PROCC), Oswaldo Cruz Foundation, Rio de Janeiro, Brazil
[3]Centre on Climate Change and Planetary Health, and [4]Centre for Mathematical Modelling of Infectious Diseases, London School of Hygiene and Tropical Medicine, London, UK
[5]Barcelona Institute for Global Health (ISGlobal), Barcelona, Spain

LPF, 0000-0001-9012-9382; OGC, 0000-0002-3289-3195; RL, 0000-0003-3939-7343; MSC, 0000-0002-9566-0284

Dengue, an arboviral disease transmitted by *Aedes* mosquitoes, has been endemic in Brazil for decades. However, vector-control strategies have not led to a significant reduction in the disease burden and have not been sufficient to prevent chikungunya and Zika entry and establishment in the country. In Rio de Janeiro city, the first Zika and chikungunya epidemics were detected between 2015 and 2016, coinciding with a dengue epidemic. Understanding the behaviour of these diseases in a triple epidemic scenario is a necessary step for devising better interventions for prevention and outbreak response. We applied scan statistics analysis to detect spatio-temporal clustering for each disease separately and for all three simultaneously. In general, clusters were not detected in the same locations and time periods, possibly owing to competition between viruses for host resources, depletion of susceptible population, different introduction times and change in behaviour of the human population (e.g. intensified vector-control activities in response to increasing cases of a particular arbovirus). Simultaneous clusters of the three diseases usually included neighbourhoods with high population density and low socioeconomic status, particularly in the North region of the city. The use of space–time cluster detection can guide intensive interventions to high-risk locations in a timely manner, to improve clinical diagnosis and management, and pinpoint vector-control measures.

## 1. Introduction

Dengue has been endemic in Brazil for more than 30 years. Since 2010, all four dengue virus (DENV) serotypes circulate in the country [1]. The first chikungunya and Zika outbreaks in Brazil were detected in 2014 and 2015, respectively, both in the northeast region. In 2016, 1.5 million dengue cases, 270 000 chikungunya cases and more than 200 000 Zika cases were notified in the country [2]. Initially described as a benign disease, Zika quickly became a serious public health problem after the association of the disease during pregnancy with congenital malformations, such as microcephaly, was discovered [3,4].

The co-circulation of DENV, chikungunya virus (CHIKV) and Zika virus (ZIKV) poses a serious public health and economic burden [5–8]. The Brazilian government has implemented dengue prevention and control measures in the form of vector-control interventions, but there is no evidence that vector-control has had a significant effect in reducing transmission in Brazil or other parts of the world [9]. The widespread presence of the vector (mainly *Aedes aegypti* but also *Aedes albopictus*), a highly mobile population and low or lack of herd immunity resulted in simultaneous and overlapping outbreaks of all three diseases, a phenomenon

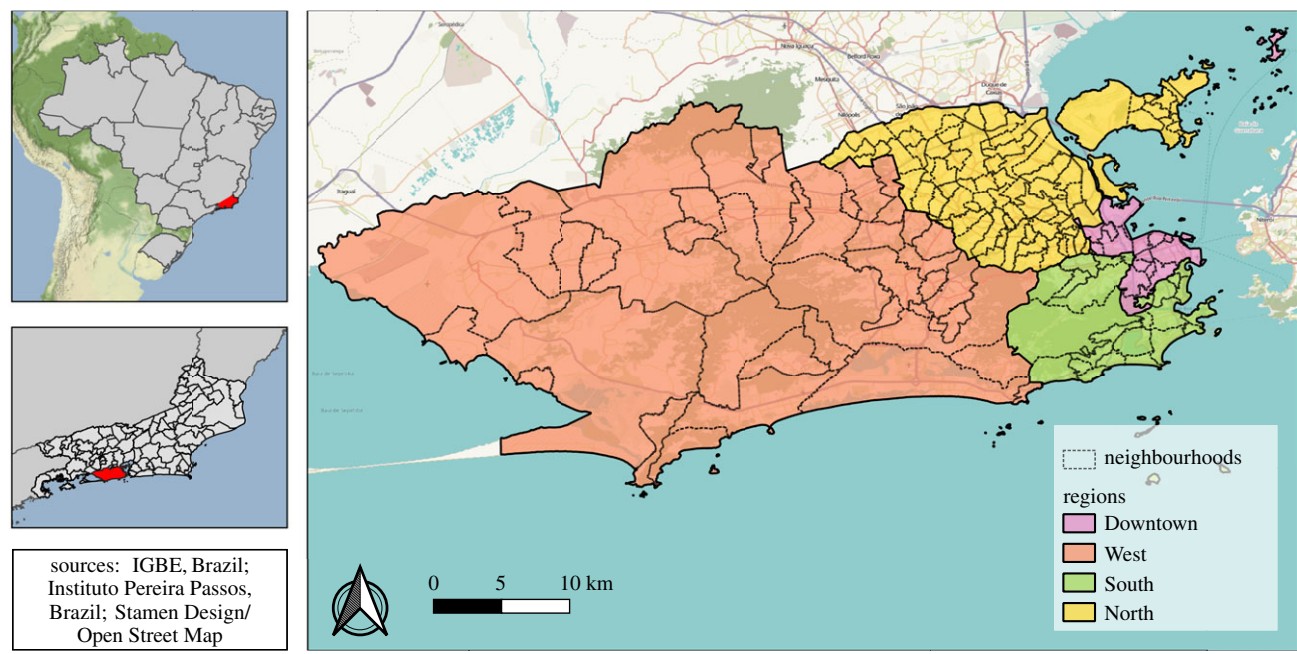

**Figure 1.** Rio de Janeiro city regions and neighbourhoods, 2010.

that has been referred to as the 'triple epidemic' [10]. In this scenario, the healthcare system needs to be prepared to account for medical interventions, which are different for each disease, and prevent severe clinical developments. Furthermore, co-infections are possible and clinical manifestations for such cases are not clear [11]. Understanding the behaviour of dengue, Zika and chikungunya, when they coexist in time and space, is a step forward in improving the design of interventions for prevention and outbreak response [12].

The Brazilian National Notifiable Diseases Information System (Sistema de Informação de Agravos de Notificação (SINAN)) is the Ministry of Health's system for surveillance of diseases included in the national list of compulsory notification. Dengue has been a notifiable disease since 1961, and chikungunya since 2011. Zika was only included in February 2016, but since June 2015, Zika was monitored through sentinel surveillance [13]. As a passive surveillance system, one of SINAN's limitations is under-reporting. However, SINAN receives a large number of notifications, and it is thought to represent the overall trend of the dengue situation in Brazil [14,15].

Considering DENV, CHIKV and ZIKV share the same vectors and human hosts, we conducted a spatio-temporal analysis of notified cases to identify clusters and understand the dynamics of these diseases in a 'triple epidemic' scenario. Rio de Janeiro was the chosen city for this analysis for the following reasons: a history of large dengue epidemics with sustained transmission; the recent occurrence of CHIKV and ZIKV epidemics in 2015–2016; co-circulation of DENV, CHIKV and ZIKV; a high number of reported cases; the possibility to work with georeferenced cases in an intra-urban context; multiple environmental settings within the city; high human mobility; vector abundance; and health professionals experienced in dealing with dengue as a result of the epidemiological scenario.

## 2. Methods

### (a) Study site

Rio de Janeiro is the second largest city in Brazil, with approximately 6.3 million inhabitants (2010 census), 1204 km² and 160 neighbourhoods (figure 1). The city has the 45th highest human development index (HDI) of the country, of 0.799 (varying from 0.604 to 0.959 inside the city) [16]. The population density is 5249 inhabitants km⁻². Population density and green areas vary across neighbourhoods (electronic supplementary material, figure S1). Rio de Janeiro has a tropical climate, with temperature and rainfall varying depending on altitude, vegetation and ocean proximity. The average annual temperature is 23.7°C, and the annual accumulated precipitation is 1069 mm [17].

The 160 neighbourhoods are grouped into four large regions (North, South, Downtown and West, figure 1), reflecting the geographical position and history of occupation. Almost all neighbourhoods are a mixture of very poor slums ('favelas') and more affluent areas of residence. The North region is very urbanized, with high population density, few green areas and very large favelas. Nearly 27% of the population of this region, almost 2.4 million people, lived in favelas in the 2010 demographic census [18]. The South region is the most popular tourist destination in Rio de Janeiro, with famous beaches, green areas and neighbourhoods with the highest HDI of the city [16]. The Downtown region is the historical, commercial and financial centre of the city, with many green areas and cultural establishments. Finally, the West region has undergone intense urbanization more recently, and is less densely populated [17].

### (b) Data

Data on dengue, chikungunya and Zika cases were obtained from SINAN via the Rio de Janeiro Municipal Secretariat of Heath, and are publicly available [19]. The Municipal Secretariat of Health georeferenced 91% of dengue cases, 95% of chikungunya cases and 92% of Zika cases, using the address of the patient's residence.

We analysed notified cases of dengue, Zika and chikungunya (confirmed by laboratory or by clinical–epidemiological criteria) occurring in Rio de Janeiro municipality between 2 August 2015 and 31 December 2016 (epidemiological weeks 31, 2015 and 52, 2016), grouped by epidemiological week and neighbourhood of residence. Case definitions follow Ministry of Health protocols [13,20,21]. Population data by neighbourhood and shapefiles were obtained from the Instituto Pereira Passos [22].

### (c) Space–time analysis

To detect spatio-temporal clusters of arboviral diseases in Rio de Janeiro, we used the Kulldorff's scan statistic. This methodology

**Table 1.** Notified cases of dengue, chikungunya and Zika between epidemiological weeks 31, 2015 and 52, 2016 in Rio de Janeiro city, Brazil.

| | dengue | chikungunya | Zika |
|---|---|---|---|
| total number of cases | 26 546 | 13 627 | 35 857 |
| incidence per 100 000 inhabitants | 420.0 | 215.6 | 567.3 |
| maximum no. of cases per week | 2118 | 1118 | 1811 |
| week with maximum no. of cases | 14, 2016 | 17, 2016 | 01, 2016 |
| no. of neighbourhoods with at least one case | 157 | 159 | 160 |
| no. of neighbourhoods with at least 10 cases | 145 | 136 | 155 |

was chosen as it (i) allows detection of space–time clusters for discrete Poisson probability distributions; (ii) tests the statistical significance and corrects for multiple testing; (iii) examines disease dynamics in continuous time; (iv) estimates the relative risk (RR) for each cluster (considering the underlying population); and (v) it can simultaneously evaluate more than one disease [23].

The scan statistic was applied for each disease individually and all three diseases simultaneously (multivariate scan statistic with multiple datasets). Through moving cylinders across space (i.e. the base of the cylinder) and time (i.e. the height of the cylinder), it identifies high-risk clusters by comparing the observed number of cases to the expected number of cases inside the cylinder [24]. In our analysis, the neighbourhood was considered as part of the cylinder if its centroid was located within the base of the cylinder. The null hypothesis is that the risk within the cylinder is equal to the one outside. For each cylinder, its expected number of cases ($E[c]$) is equal to the total number of cases in the city ($C$) divided by the total city population ($P$), times the population within the cylinder ($p$) [23]:

$$E[c] = \frac{C}{P} \times p. \tag{2.1}$$

The detected clusters are ordered in the Results section according to the log-likelihood ratio (LLR), such that the cluster with the maximum LLR is the most likely cluster, that is, the cluster least likely to be due to chance. The LLR is calculated using the following equation [23]:

$$LLR = \left(\frac{c}{E[c]}\right)^c \left(\frac{C-c}{C-E[c]}\right)^{C-c} I(), \tag{2.2}$$

where $c$ is the number of cases inside the cluster and $I()$ is an indicator function that is equal to 1 when the cylinder has more cases than expected and 0 otherwise. The RR for each cluster is calculated using the following equation [23]:

$$RR = \frac{c/E[c]}{(C-c)/(C-E[c])}. \tag{2.3}$$

The multivariate scan statistic for multiple datasets was applied to simultaneously search for clusters of dengue, Zika and chikungunya that coincided in time and space. This technique calculates for each cluster the LLR for each disease. Then, the LLR for a particular cluster is calculated as the sum of the LLR for the three diseases. As for a single disease, the maximum of the sum of the LLRs constitutes the most likely cluster [23,25].

For each model, Monte Carlo simulations ($n = 999$) were performed to assess statistical significance. We considered statistically significant clusters ($p < 0.05$) with no geographical overlap and that included a maximum of 50% of the city's population (nearly 3.1 million people). After testing several combinations of temporal and spatial parameters, we chose the combination that resulted in a reasonable number of clusters

that could be targeted for local interventions (electronic supplementary material, figure S2). The temporal window was set to be at least one week and a maximum of four weeks. Clusters were restricted to have at least five cases and, in the output parameters, to include a maximum of 5% of the city's population (nearly 315 000 people).

SaTScan™ (v. 9.5) software was applied within R (v. 3.4.4), using the package rsatscan (v. 0.3.9200) [26–28]. The R code is available at https://github.com/laispfreitas/satscan_dzc/blob/master/script_satscan_dzc_rio [29]. Maps were produced using QGIS (v. 3.8.1) and ggplot2 (v. 3.1.0) package in R [30,31].

## 3. Results

In Rio de Janeiro, between 2 August 2015 and 31 December 2016 (epidemiological weeks 31, 2015 and 52, 2016), 76 030 cases of dengue, chikungunya and Zika were reported (table 1). More than 85% of neighbourhoods had at least 10 cases of each disease. Zika presented the highest number of notifications, resulting in an incidence of 567.3 cases per 100 000 inhabitants. Most cases occurred between December 2015 and June 2016 (86.2%). The epidemic curves differed slightly in time, with high incidence of all three diseases between April and June 2016 (figure 2). In March 2016, Zika cases started to decrease, while dengue and chikungunya cases were still on the increase. While dengue and Zika were active by the end of 2015, chikungunya cases only started to rise in March 2016. Notifications of the three diseases declined after May.

### (a) Dengue cases clusters
Scan statistics detected 18 dengue cases clusters in different parts of the city (figure 3a). The most likely cluster was located in the North region of Rio de Janeiro city. Cluster 2 contained only one neighbourhood in the Downtown area and presented the highest RR, of 151.90 (electronic supplementary material, table S1 and figure S3A). Clusters were detected within a short time period, from March to May 2016, except for cluster 16 that started in December 2015 (figure 3b). The first dengue cluster in time included neighbourhoods located between the South and the West regions (electronic supplementary material, figure S4A).

### (b) Chikungunya cases clusters
For chikungunya, 15 clusters were detected (figure 4a). Unlike dengue, chikungunya clusters were seen less frequently in the West region. The most likely cluster was located in the Downtown of Rio de Janeiro city and had the highest RR, of 25.77 (electronic supplementary material, table S2 and figure S3B). Clusters were also detected within

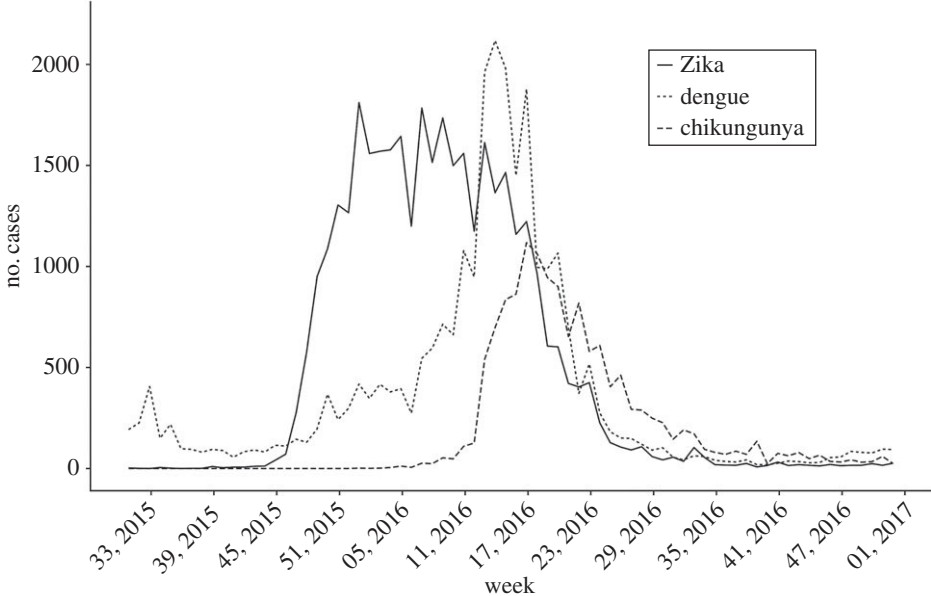

**Figure 2.** Number of reported dengue (dotted line), chikungunya (dashed line) and Zika (solid line) cases between 2 August 2015 and 31 December 2016, Rio de Janeiro city, Brazil.

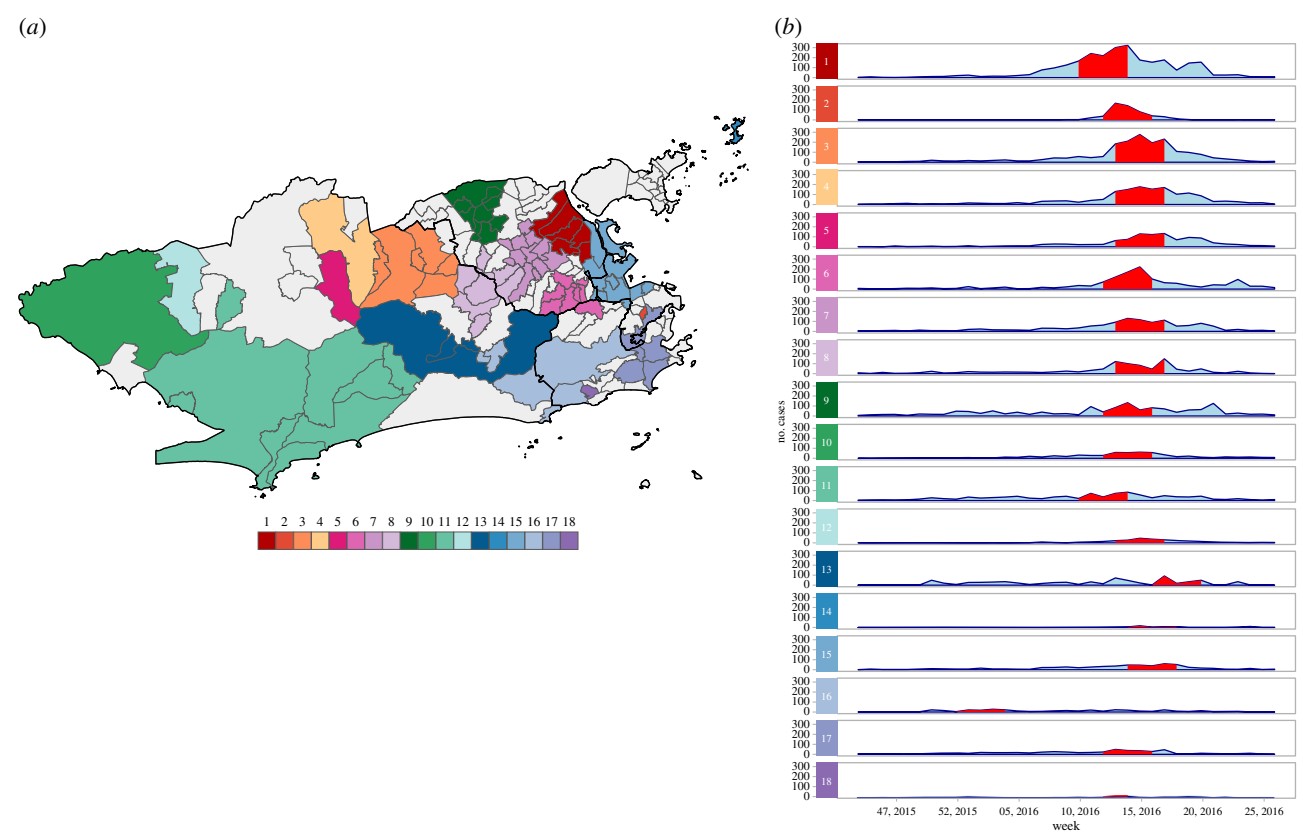

**Figure 3.** (*a*) Dengue cases clusters and (*b*) temporal distribution of dengue cases by cluster, between epidemiological weeks 31, 2015 and 52, 2016, Rio de Janeiro city, Brazil. Red bands represent the time period at which the cluster was detected. Clusters are ordered according to the maximum LLR, with 1 being the most likely cluster.

a restricted time period, between 20 March and 11 June (figure 4*b*). The first chikungunya cluster in time occurred in the northern border of the city (electronic supplementary material, figure S4B).

## (c) Zika cases clusters

There were 15 Zika clusters, distributed all over the city, similar to the observed pattern for dengue (figure 5*a*). The most likely cluster was located in the West of Rio de Janeiro city,

a region where chikungunya clusters were rarely observed. This cluster also had the highest RR, of 13.57 (electronic supplementary material, table S3 and figure S3C). In contrast with dengue and chikungunya, Zika clusters occurred over a longer period of time, between the end of November 2015 and May 2016 (figure 5*b*). The third most likely cluster occurred eight weeks after the first one. The first Zika clusters in time emerged in the North of the city (electronic supplementary material, figure S4C).

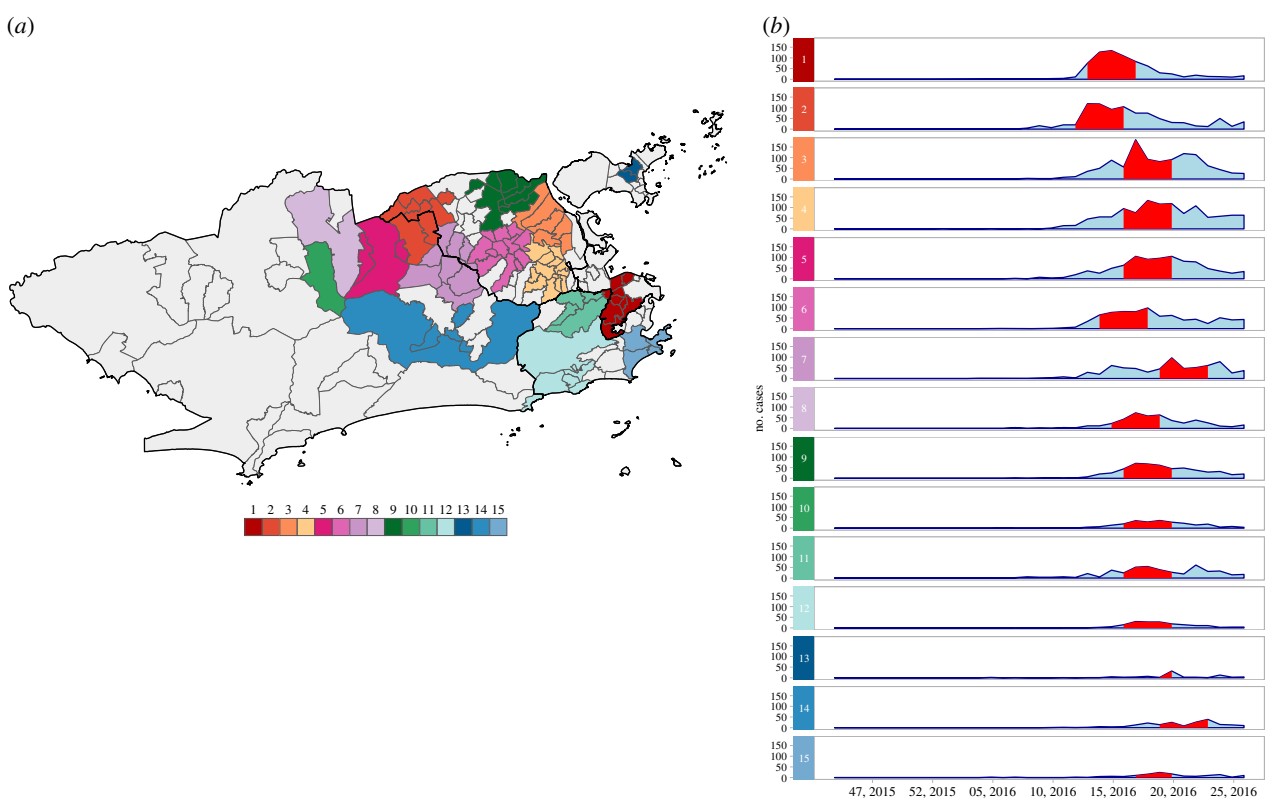

**Figure 4.** (*a*) Chikungunya cases clusters and (*b*) temporal distribution of chikungunya cases by cluster, between epidemiological weeks 31, 2015 and 52, 2016, Rio de Janeiro city, Brazil. Red bands represent the time period at which the cluster was detected. Clusters are ordered according to the maximum LLR, with 1 being the most likely cluster.

## (d) Dengue, chikungunya and Zika multivariate clusters

The multivariate scan statistic for multiple datasets detected 16 clusters, of which nine showed dengue, chikungunya and Zika occurring simultaneously; five showed overlapping dengue and Zika outbreaks and two showed only outbreaks of Zika (figure 6). The most likely cluster was predominantly located in the Downtown region of the city and had the highest RRs for dengue (21.16), chikungunya (25.30) and Zika (7.66) among the simultaneous clusters for the three diseases (electronic supplementary material, table S4).

Of the 160 neighbourhoods assessed, 56 (35.0%) had clusters for the three diseases simultaneously. Of the nine simultaneous clusters, five were located in the North of the city, three in the West and one in the Downtown.

## 4. Discussion

This is, to our knowledge, the first study exploring space–time clustering of dengue, chikungunya and Zika in an intra-urban region. The data analysed are rare and of great value, as they include triple epidemics with a large number of cases. Also, this study included the first ever epidemics of chikungunya and Zika in Rio de Janeiro city. In brief, detected clusters for each disease presented different dynamics in time and space. Dengue and Zika clusters were found across the city, with Zika clusters persisting over a longer time period. Chikungunya clusters were more concentrated in the North and Downtown regions. Simultaneous clusters of the three diseases were more likely in neighbourhoods with a combination of high population density and low socioeconomic status.

Dengue, chikungunya and Zika cases were notified across the whole city. The epidemic curves varied slightly in time,

reaching maximum numbers in different weeks. The number of cases of the three diseases declined after May, coinciding with the end of the rainy and warm season (electronic supplementary material, figure S5). This reflects the vectors ecology, as *Ae. aegypti* and *Ae. albopictus* breed in pools of water, and temperatures of around 25–30°C accelerate the reproductive cycle and increase infectivity and transmissibility [32].

The simultaneous decrease in Zika and increase in chikungunya cases was also observed in a study in Recife, northeast Brazil, and in a study analysing laboratory-confirmed cases in the state of Rio de Janeiro [33,34]. The authors from both studies interpreted this as a displacement of Zika caused by chikungunya. In Rio de Janeiro city, CHIKV was already circulating at the beginning of 2016 but did not trigger an epidemic before Zika cases started decreasing (which was possibly caused by the depletion of ZIKV susceptible hosts). We hypothesize that ZIKV circulation could be inhibiting CHIKV, rather than CHIKV introduction displacing ZIKV. When simultaneously co-infected with both viruses, *Ae. aegypti* was found to transmit ZIKV at a higher rate than CHIKV [35]. The transmission rates for simultaneous co-infection were not significantly different from the rates for single infection. However, it is not clear how the viruses interact when the mosquito is infected sequentially, not simultaneously. That is, when the mosquito is infected by one virus after biting one person and later by another virus by biting another person, the most likely scenario in nature considering co-infections in humans are not common [11]. Under specific laboratory conditions, sequential infection with CHIKV and ZIKV led to enhanced ZIKV transmission [36]. It is also possible that at the beginning of 2016, the prevalence of CHIKV was too low to trigger an epidemic, and that the virus was subsequently reintroduced to the city.

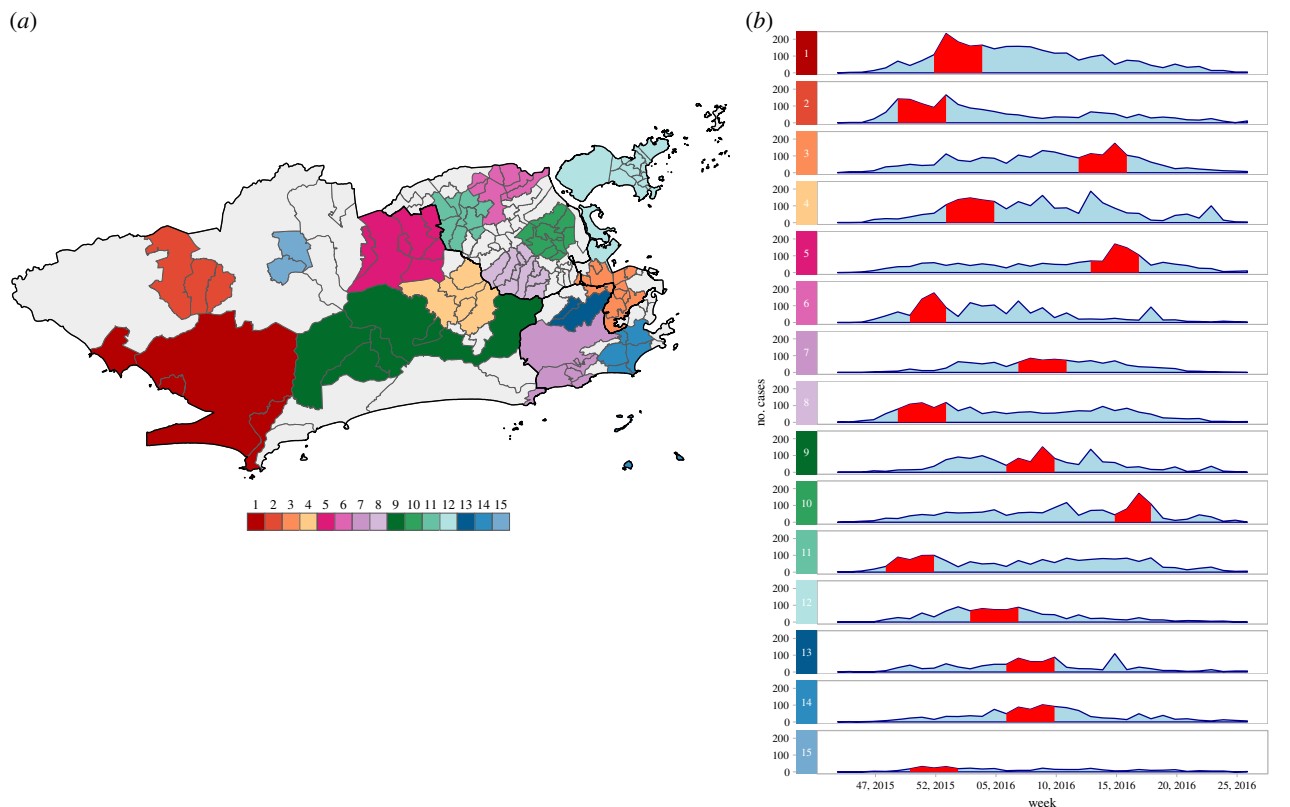

**Figure 5.** (a) Zika cases clusters and (b) temporal distribution of Zika cases by cluster, between epidemiological weeks 31, 2015 and 52, 2016, Rio de Janeiro city, Brazil. Red bands represent the time period at which the cluster was detected. Clusters are ordered according to the maximum LLR, with 1 being the most likely cluster.

Scan analysis successfully identified clusters of dengue, chikungunya and Zika. This method has been used to identify risk areas for arboviral diseases in other locations [37–40]. One of the advantages of this method over commonly used exploratory methods is that it looks for clusters in time continuously, accounting for temporal dependency, instead of fixed and arbitrary time windows. It also tests for statistical significance, corrects for multiple testing and estimates the RR. The visual and exploratory analysis depends on subjective evaluation, whereas scan statistic methodology is more statistically robust. SATSCAN™ is a free and user-friendly tool, which could serve as a valuable disease surveillance tool, particularly in resource-limited settings [41,42]. However, the method has some limitations. Scan analysis was not designed to understand disease trajectory but can be useful in planning interventions. Also, the method detects circular clusters only, rather than clusters of irregular shapes. Therefore, if a neighbourhood with low risk of the disease is surrounded by neighbourhoods with high risk, it could be considered as part of the cluster. This can be reduced by limiting the size of the clusters.

The most likely cluster for each disease occurred in a different part of the city and in a different time period (dengue: North region, epidemiological weeks (EWs) 10–14/2016; chikungunya: Downtown region, EWs 13–17/2016; Zika: West region, EWs 52/2015–4/2016). Unlike for dengue and Zika, chikungunya clusters were not widely detected in the West of Rio de Janeiro, probably because the rainy and warm season ended before the disease could reach this region with a sufficient transmission rate to form clusters.

Zika clusters were detected over a longer period of time compared to dengue and chikungunya clusters. This could be a result of the population being naive combined with the ZIKV advantage in competing for *Ae. aegypti* mosquitoes: the *Ae. aegypti* has been described as a more efficient vector for ZIKV transmission than for DENV or CHIKV, even when co-infected [35,43]. Not only does *Ae. aegypti* transmit ZIKV at a higher rate, but it is also more easily infected by ZIKV compared to DENV and CHIKV [35].

A previous study suggested that a Zika epidemic would prevent a subsequent dengue epidemic as a consequence of cross-immunity [44]. In our study, the number of dengue cases increased after the maximum number of Zika cases. Additionally, some locations with Zika clusters also experienced dengue clusters afterwards. Like DENV, ZIKV is a flavivirus, and the structural similarity between them results in cross-immunity [45]. Whether this cross-immunity leads to disease enhancement, protection or neither is still under debate [46]. Two recent papers showing results from cohort studies shed some light upon this matter. In a paediatric cohort in Nicaragua, prior DENV infection was associated with lower rates of symptomatic Zika [47], and in a cohort in Pau da Lima, northeast Brazil, the titres of anti-DENV antibodies before the Zika epidemic were inversely associated with the risk of ZIKV infection [48]. After the epidemic of congenital Zika syndrome in Brazil was detected, many researchers questioned if it was related to the mother's anti-DENV antibodies [49,50]. There is insufficient evidence to understand the consequences of previous DENV exposure on Zika outcomes during pregnancy. However, considering the severe consequences of congenital Zika syndrome, disease surveillance using spatio-temporal scan statistics should be considered to identify high-risk areas for Zika in a timely manner and to direct preventive measures to the most at risk areas.

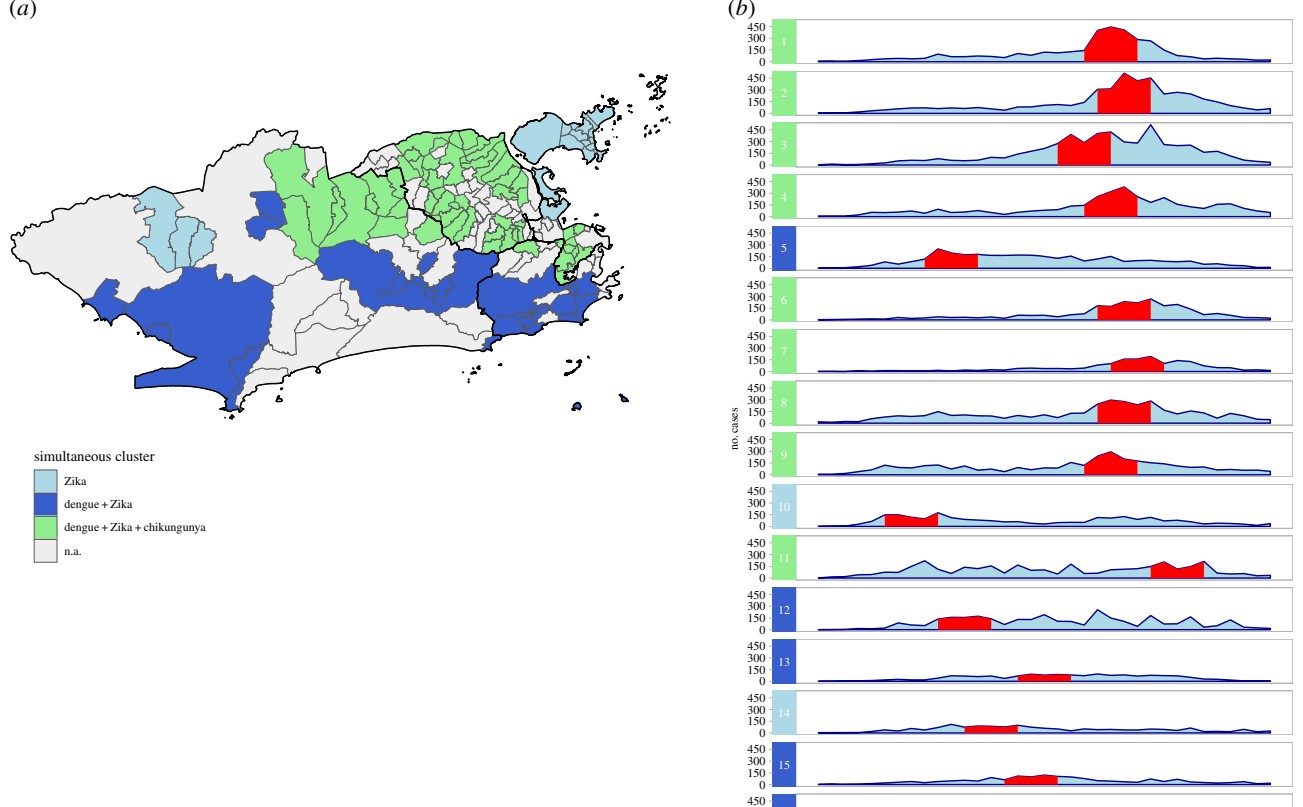

**Figure 6.** (*a*) Clusters of dengue, chikungunya and Zika detected using the multivariate scan statistic and (*b*) temporal distribution of cases by cluster, between epidemiological weeks 31, 2015 and 52, 2016, Rio de Janeiro city, Brazil. Red bands represent the time period in which the cluster was detected. Clusters are ordered according to the maximum LLR, with 1 being the most likely cluster.

Dengue, chikungunya and Zika clusters detected in Rio de Janeiro did not usually coincide in time and space, contrasting with a study in Mexico that found strong spatio-temporal coherence in the distribution of the three diseases [12]. In addition to virus interactions and competition for the resources for replication inside the vector, behaviour changes among the human population may also help explain the spatio-temporal differences in the distribution of the three diseases. A rise in the number of cases of one arboviral disease may promote vector-control activities, which in turn may decrease the number of cases and hinder the establishment of another arbovirus in that location [51]. Also, wealthier areas may have better vector-control interventions, another factor that may lead to different spatial distributions. Alternatively, the observed differences may be a result of different introduction times of the viruses across the city. In a previous study conducted in a large city of Bahia state, Brazil, the chikungunya epidemic curve showed a temporal displacement only in the first wave, synchronizing with the dengue curve in the second wave [52]. We only analysed the first epidemic waves for chikungunya and Zika. Further investigations including subsequent years are important to elucidate if the spatio-temporal distribution of the three diseases changes after epidemic establishment.

Neighbourhoods in the North of the city were more likely to have simultaneous clusters of dengue, Zika and chikungunya, highlighting these areas as priority targets for interventions, especially the timely allocation of resources to local health services, which can become overloaded, and training of medical teams on the differential diagnosis

between the diseases. The preparedness of the health service is also important considering co-infections are possible and clinical outcomes are not clear for such cases [11,53]. Simultaneous clusters also suggest increased exposure to *Ae. aegypti* and, therefore, vector-control activities should also be intensified in these locations.

This study captured the first ever-reported cases of chikungunya in the city, pinpointing its source in the North of the city. Note, dengue has been endemic in Rio de Janeiro for the last three decades and notification of Zika cases was only established in the municipality in October 2015 (after the Zika epidemic had already begun). The North of Rio de Janeiro has already been identified as a hot spot for dengue and as a key region for dengue diffusion [54,55]. Such studies also identified Catumbi, a neighbourhood in the Downtown area, as a high-risk location for dengue. In our findings, Catumbi comprised the most likely chikungunya cluster, the second most likely cluster for dengue and the third most likely for Zika. Additionally, the clusters in Catumbi coincided in time (most likely cluster in the multivariate scan analysis). Further investigations should be conducted to understand why this neighbourhood in particular is a high-risk location for arboviruses.

The North of the city is marked by a combination of high population density and a lower HDI than the city average [16]. The high population density facilitates the mosquito–human contact and hence the chance of becoming infected. In Rio de Janeiro, areas in or near favelas were detected as hot spots for dengue [55]. Consistent with our findings, a study conducted in French Guiana indicated that, early in

the epidemic, the poorest neighbourhoods would have a greater risk for CHIKV infection [56]. In the first dengue epidemic in a city of São Paulo state, Brazil, authors found a direct relationship between low socioeconomic conditions and dengue [57]. We did not observe this relationship for dengue possibly because dengue has already had sustained transmission in the city for decades. The link between poverty and arbovirus is controversial [58]. Nonetheless, locations with social and economic vulnerability more likely have poorer sanitary conditions and less efficient vector-control interventions, which would facilitate mosquito proliferation.

Some limitations affect this study. As our study population included only notified cases (i.e. only patients who sought medical care), asymptomatic cases were not captured. Mild cases are usually poorly captured by SINAN, but considering the disease awareness around Zika, people (especially women) were expected to be more concerned about seeking medical care in the case of suspected Zika. As Zika, dengue and chikungunya share some symptoms, the disease awareness may have boosted the notification of mild cases of the three diseases. The similar clinical manifestations of dengue, Zika and chikungunya also represent a limitation. This limitation is inherent of every study using notified cases, as only a small proportion of cases are laboratory confirmed (8.5%, 30.4% and 4.1%, for dengue, chikungunya and Zika, respectively, in our dataset). Also, we did not have information on co-infections within the disease surveillance database. However, as co-infections are rare, this should not have affected our analysis. In a national survey in Colombia, co-infections accounted for 0.14% of the arboviral diseases cases [59].

A small percentage of cases (8%) that were not georeferenced (and hence, not included in this study) could potentially result in a selection bias. It is possible that cases occurring in favelas, where addresses are sometimes not standardized, have a higher chance of not being georeferenced. Clustering was based on the neighbourhood of residence only, yet infection can happen at other places, such as the workplace.

Vector-control strategies have not been effective in abating dengue or in preventing the entry of Zika and chikungunya in Rio de Janeiro. The identification of clusters in space and time allows actions to be intensified in high-risk locations in a timely manner. It is essential that healthcare facilities are prepared to prevent severe clinical developments (such as haemorrhagic dengue fever, chronic pain among chikungunya cases and congenital Zika syndrome) and deaths. Special attention should be given to neighbourhoods with high population density and low socioeconomic status. As vector-control relies on community participation, it is important to enhance community engagement and build trust among all members of the community. People living in neighbourhoods with poor sanitation and a low development index may be less likely to adhere and to maintain prevention activities. Measures to reduce inequity should be accompanied by sustained community engagement [51]. Finally, we suggest the implementation of spatio-temporal scan statistics in the municipal surveillance routine as a tool to optimize prevention strategies.

Ethics. This study was approved by the Research Ethics Committee of Escola Nacional de Saúde Pública Sergio Arouca (ENSP)—Oswaldo Cruz Foundation, approval number 2.879.430. Informed consent was not required as this is a study using secondary data.

Data accessibility. The data underlying the results presented in the study are from the Rio de Janeiro Municipal Secretariat of Health. The data can be downloaded at http://www.rio.rj.gov.br/web/sms/exibeConteudo?id=6525201. The code was made available at https://github.com/laispfreitas/satscan_dzc/blob/master/script_satscan_dzc_rio [29].

Authors' contributions. L.P.F., O.G.C. and M.S.C. conceived the study; L.P.F. and O.G.C. carried out the statistical analysis; all authors contributed to the interpretation of the data. L.P.F. and M.S.C. drafted the manuscript; O.G.C. and R.L. critically revised the manuscript. All authors read and approved the final manuscript.

Competing interests. We declare we have no competing interests.

Funding. This work was supported in part by the Coordenação de Aperfeiçoamento de Pessoal de Nível Superior—Brasil (CAPES, http://www.capes.gov.br/)—Finance Code 001, to L.P.F. M.S.C. received grants from Fundação Carlos Chagas Filho de Amparo à Pesquisa do Estado do Rio de Janeiro (FAPERJ, http://www.faperj.br/, grant no. E_26/201.356/2014) and support from Conselho Nacional de Desenvolvimento Científico e Tecnológico (CNPq, http://www.cnpq.br/, grant no. 304101/2017-6). R.L. was funded by a Royal Society Dorothy Hodgkin Fellowship (https://royalsociety.org). The funders did not influence the content of this manuscript nor the decision to submit it for publication.

Acknowledgements. The authors would like to thank the Municipal Secretariat of Health for providing the data on reported cases, and Dr Reinaldo Souza dos Santos (Escola Nacional de Saúde Pública Sergio Arouca) and Dr Valéria Saraceni (Municipal Secretary of Health and Civil Defense, City Hall of Rio de Janeiro) for reviewing and providing helpful feedback.

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
