## [Reviewer comments · Proceedings of the Royal Society B: Biological Sciences]

Review History

RSPB-2019-0310.R0 (Original submission)

Review form: Reviewer 1

Recommendation

Accept with minor revision (please list in comments)

Scientific importance: Is the manuscript an original and important contribution to its field?

Acceptable

General interest: Is the paper of sufficient general interest?

Good

Quality of the paper: Is the overall quality of the paper suitable?

Good

Is the length of the paper justified?

Yes

Should the paper be seen by a specialist statistical reviewer?

Yes

Do you have any concerns about statistical analyses in this paper? If so, please specify them explicitly in your report.

No

It is a condition of publication that authors make their supporting data, code and materials available - either as supplementary material or hosted in an external repository. Please rate, if applicable, the supporting data on the following criteria.

Is it accessible?

Yes

Is it clear?

Yes

Is it adequate?

Yes

Do you have any ethical concerns with this paper?

No

Comments to the Author

Freitas et al. present an interesting spatio-temporal analysis of an important triple-epidemic. They provide a generally clear exposition on the use of spatial scan statistics, and raise a number of important points which must be considered for coincident epidemics of these three arboviruses.

We have quite a few, but all relatively minor, comments:

General:

Would you consider making the R code available in an online repository?

Introduction:

Lines 43-44: Whilst the statement regarding the public health & economic burden is probably true, the references given, which focus on individual diseases and are not economic analyses, don't seem relevant to the statement. Are there more relevant ones?

Line 29: Could you consider defining the term 'social vulnerability'?

Methods:

The right panel in figure 1 is hard to read as there are too many things going on in it (its especially hard to tell the population density). Would you be able to make it clearer - perhaps by doing two maps (one for neighborhoods, and one for density)

line 94: could you reference the URL for the data here.

The section about space-time analysis is currently a bit vague - could you please expand a little on the details (perhaps in the supplementary material if there is not room in the main text). In particular:

How did you calculate/define the expected number of cases?

How did you calculate the likelihood ratio for a cluster?

Line 108: should these be the most likely cluster with more observed than expected cases, as in the multivariate situation? (Apologies if this is a misunderstanding!)

line 109-112 is quite confusing. Could you use an equation instead?

Also, why is the risk denominator the expected cases, rather than the population - or is the

population proportional to the expected cases in this model?

It looks like the clusters in supp. figure 1 have some overlap, but you state that the clusters should have no spatial overlap in line 121. Please clarify this.

Results:

In the caption for figures 3-5, please explain what the red bands in panel B mean (presumably the time window?). Also, please mention in the caption that clusters are numbered in order of likelihood.

Perhaps consider reporting the p-values of the clusters in tables 1-5

In the supp figure 3, it is not clear which cluster came first. Could you consider labeling the cluster which came first, using a clearer color scale, or using a different color scale for A, B and C? Also, it might help with interpretation to include the region (N, S, W, E) borders on these figures.

Discussion:

Perhaps it would be good to discuss, generally, some of the limitations of spatial scan statistics, to aid readers not familiar with the methodology.

The recent paper by Gordon et al.

(<https://journals.plos.org/plosmedicine/article?id=10.1371/journal.pmed.1002726>, no affiliation with the reviewers) might be relevant to your discussion of dengue-Zika cross-immunity.

The claim in lines 248-250, that herd immunity probably did not have a significant impact on the Zika or dengue dynamics, seems too strong based on the data and references presented. For instance, there is no consideration of the counterfactual situation (of no/lower dengue seroprevalence), or of transmissibility of the viruses. Perhaps remove this claim, or support it. Lines 255-257 please could you provide references for the claims that (a) many researchers questioned if CZS was related to maternal DENV antibodies, and (b) that there is insufficient evidence for this claim.

Line 283: could you provide reference(s) for the claim that the link between poverty and arboviruses is controversial?

Line 300: this statement about misdiagnosis is a little unclear – what exactly are the differences you refer to between? Also, isn't this statement only true if the misdiagnosis is unbiased? For instance, how would misdiagnosis in a particular direction (e.g. Chik often misdiagnosed as Zika, but not vice versa?) affect interpretation? In general, the discussion around misdiagnosis could perhaps be expanded on a little.

Could you perhaps discuss how the detection of joint clusters might differentially affect control, compared to control for single-disease clusters (if you think there would be a difference)?

Is it likely that some of the clusters are spurious? For instance if all of the p-values are close to 0.05 (you have more than 20 clusters).

Review form: Reviewer 2 (J Lourenço)

Recommendation

Major revision is needed (please make suggestions in comments)

Scientific importance: Is the manuscript an original and important contribution to its field?

Marginal

General interest: Is the paper of sufficient general interest?

Marginal

Quality of the paper: Is the overall quality of the paper suitable?

Acceptable

Is the length of the paper justified?

Yes

Should the paper be seen by a specialist statistical reviewer?

No

Do you have any concerns about statistical analyses in this paper? If so, please specify them explicitly in your report.

No

It is a condition of publication that authors make their supporting data, code and materials available - either as supplementary material or hosted in an external repository. Please rate, if applicable, the supporting data on the following criteria.

Is it accessible?

No

Is it clear?

Yes

Is it adequate?

No

Do you have any ethical concerns with this paper?

No

Comments to the Author

In this study, the authors explore the geo-temporal patterns of Zika, dengue and chikungunya viruses in Rio de Janeiro, Brazil. The text is well written, with good visualisations of results and description of methods.

My main concerns about this study are (1) the dependence of results interpretation on strong competition between Zika, dengue and chikungunya viruses, (2) as a reader, failing to understand why the method used is ideal to the results presented, and (3) whether the results are innovative or enough for publication in this journal.

(1) I recognize that competition is a topic of debate (whatever the mechanism may be), but as far as I am aware there isn't enough literature to support this as the main driver of the patterns observed. Even if the authors, upon revision, can find enough literature support (which may be possible), the biggest fault at the moment is that alternative / complementary hypothesis / mechanisms are not discussed in the text - for example, different introduction times are the parsimonious explanation. I recommend the authors discuss these alternatives in their manuscript, as do the authors of the few references included in the current version (see my comments below).

(2) Although the methods are well described, I don't think that the manuscript allows the reader to understand why scan statistics are ideal for this analyses, or why the data requires that method versus a different one (see my comments below).

(3) The results are clear, but the manuscript is essentially a large discussion. My worry is that a small set of results, which overlap with previous literature, may not be enough for this journal. I will nonetheless suggest a major revision and will let the Editor decide on this topic.

"Data" section

Needs a bit more detail. For instance, is geolocation by residence or hospital / medical centre? (later in the discussion is mentioned as residence, but such information should be placed here).

Also, are case counts suspected? confirmed? What are the case definitions in SINAN?

I recognize that the authors have stated the origin of the data, but unless SINAN has very strict publication rules, such data should be made available with this manuscript (for reproducibility purposes, for example).

"Space-time analysis" section

In line 110, what do the authors mean by "expected number of cases within the cluster"?

In line 113, "search" should read "searching".

In line 115, the word 'window' is introduced for the first time - what type of window? time?

The scan statistic approach is based on 'circles' to detect clusters (as stated by the authors). This section should describe how the authors decide if the circle includes a certain region, given that regions are "irregular shapes" (as termed by the authors). How much of an area needs to be included in the circle to be considered part of the cluster? and how is this dealt in light of the fact that different regions have very different areas; and such areas actually seem to have very strong correlation from west to east?

"Results" section

Line 144: why is it interesting that the Zika epidemic does not have a clear peak?

Line 156: "The first dengue cluster in time was detected in the West zone (Supplementary material Figure 3A)." Maybe an misinterpreting the fig S3A, but it seems to show the opposite of what is stated? To show that the first dengue case was in the west, shouldn't there be yellow regions in the West?

At the end of the results I am left without being convinced why the scan statistics method is valuable. That is, it is clear what the method does from the text included, but what this method adds, or what its advantages are in relation to other possible methods is difficult to evaluate. For instance, had the authors simply looked at clusters as groups of adjacent regions (not within circles of a certain radius) with cases in certain time windows, would this have resulted in exactly the same results? This is not a criticism of the method used, but I expect other readers to have the same doubt. It is important for the authors to discuss this particular topic at some point in the text.

"Discussion" section

line 217: "The number of cases of the three diseases declined after May, coinciding with the end of the rainy and warm season" - this is an opportunity to present such pattern. I suggest the authors include a supplementary figure.

line 220: The authors state: "In a study in Recife, Northeast Brazil, the simultaneous decrease of Zika and increase of chikungunya cases was also observed. The authors interpreted this as a

displacement of Zika caused by chikungunya [18]. For Rio de Janeiro city, this might not be the case, as CHIKV caused only a few cases at beginning of 2016, and only started to rise when Zika cases decreased (the depletion of susceptible hosts). Therefore, we hypothesise that ZIKV circulation inhibited CHIKV, rather than CHIKV introduction displacing ZIKV." The argument of competition between these arboviruses is therefore an essential part of this manuscript. However, I am unsure that this argument is sufficiently supported by the results presented - although competition may be one of the possible solutions. The reference given by the authors ([18]) refers to alternative explanations in Discussion: "(...) These data suggest the displacement of ZIKV by CHIKV in the study area, possibly caused by virus competition in humans and mosquito vectors, and other factors such as acquired immunity to ZIKV in the human population and the high transmission efficiency of CHIKV. Displacement patterns have been observed for distinct DENV serotypes in endemic areas [31, 32] and may occur with distinct arboviruses sharing the same hosts." It is important to note that the example of DENV in this statement should not apply (in my opinion) to the other arboviruses - it is true that DENV serotypes present displacement patterns, but this is not a consistent observation for the others. Indeed out-of-phase epidemics of these arboviruses is an apparent phenomenon of first epidemics only (see for instance [d]). The authors in [18] further state: "Reported rates of ZIKV/CHIKV co-infections in humans are, in general, low, ranging from 0% to 4.6% [33, 35, 36]. Interestingly, *Ae. aegypti* mosquitoes that are co-infected with ZIKV and CHIKV are capable of transmitting both viruses [37, 38], contradicting in a way the idea of competition. However, viral load of ZIKV in mammalian cells, mosquito cells and whole mosquitoes decrease upon co-infection with ZIKV and CHIKV [37, 38]." Note that alternative and complementary hypothesis / factors are given. In the current manuscript, this is not the case. Finally, it is critical to discuss and consider that the timings, shapes and peaks of these 3 arboviruses are more parsimoniously explained by time of introduction and herd-immunity than they are by competition. If competition was the main biological mechanism dictating the observed patterns, then epidemics in the following years should also present displacement patterns - as mentioned above, this seems not to be the case in most regions. I think this is a critical topic for this manuscript which needs to be better discussed and supported.

line 235: "Not only does *Ae. aegypti* transmit ZIKV at a higher rate, but it is also more easily infected by ZIKV compared to DENV and CHIKV" - this statement should have a citation.

line 240: "Further studies are needed to understand the importance of *Ae. albopictus* in CHIKV transmission." - should this statement include also ZIKV?

line 246: "In our study, the number of dengue cases increased after the peak of Zika cases" - In previous sections the authors refer to the interesting observation that Zika had no peak. This statement should be changed.

line 260: "Dengue, chikungunya, and Zika clusters detected in Rio de Janeiro do not usually coincided in time and space" - this statement needs a citation.

line 263: "In addition to virus interactions and competition for the resources for replication inside the vector, behaviour changes may also impact disease dynamics. A rise in the number of cases may promote vector-control activities, which in turn may decrease the number of cases and hinder the establishment of another arbovirus [25]." - While host behaviour may be one possible driver of this observation, it is again important to offer complementary or alternative hypotheses. For instance, as observed in other studies, is it instead parsimonious that climate dictates or strongly influences the synced end of all 3 epidemics?

line 270: "As dengue has been endemic in Rio de Janeiro for the last three decades and notification of Zika cases was only established in the municipality in October 2015, it was only

possible to detect the first disease cluster for chikungunya and pinpoint its source in the North of the city, highlighting once again the importance of interventions in this area." - I find this sentence difficult and am unsure what the intention of the authors is. Please rephrase.

line 273: "The North of Rio de Janeiro has already been identified as a hot spot for dengue and as a key region for dengue diffusion." - this needs a citation.

line 283: "The link between poverty and arbovirus is controversial." - while it may be controversial, many studies have found this relationship with statistical significance. The authors could contribute to make this issue less controversial by citing such studies and spread their findings.

line 293: "Mild cases usually are poorly" should read "are usually"

line 301: "In addition, the extensive experience of health care professionals working in Rio de Janeiro, in detecting and diagnosing dengue symptoms, is thought to reduce the probability of misdiagnosis." - needs citation

Other comments

line 20: "and were not sufficient" should be "have not been sufficient"

lines 22, 23: "Understanding the behaviour of these diseases in a triple epidemic scenario is a necessary step for devising better interventions" - why is it a necessary step?

line 49: "a phenomenon that has been referred to as the 'triple epidemic'" - where has this been shown to be referred this way? please cite.

line 50: "Understanding the behaviour of dengue, Zika, and chikungunya, when they compete in time and space, is a step forward in improving the design of interventions for prevention and outbreak response" - this is one of the statements that assumed the role of competition but that is not really sustained by the text or results in the manuscript.

line 58: "SINAN receives a large number of notifications and it thought to accurately represent the overall trend of the dengue situation in Brazil" - There reference given is in Portuguese and this may be insufficient for a general audience. Plus, it is also the case that other studies have suggested that SINAN (as most passive surveillance systems) does not 'accurately' capture dengue's trends (e.g. [a]). The authors confirm this in the discussion by dwelling around asymptomatic cases and clinical overlap with other viruses.

line 78: What about minimum temperatures? and humidity? these have been shown to be important (e.g. [b,c]) and the authors have an opportunity there to describe the local scenario.

line 244: "in cross-immunity. [23] Whether" - should have [23] before '.

References:

[a] Silva, M. M. O., Rodrigues, M. S., Paploski, I. A. D., Kikuti, M., Kasper, A. M., Cruz, J. S., ... Ribeiro, G. S. (2016). Accuracy of dengue reporting by national surveillance system, Brazil. *Emerging Infectious Diseases*, 22, 336-339.

[b] Lourenço, J., M. Recker, 2014. The 2012 Madeira dengue outbreak: epidemiological determinants and future epidemic potential. *PLoS neglected tropical diseases* 8:e3083

[c] Alto, B. W. and S. A. Juliano, 2001. Precipitation and temperature effects on populations of *Aedes albopictus* (Diptera: Culicidae): Implications for range expansion. *Journal of Medical Entomology* 38:646–656.

[d] Faria et al 2016. Epidemiology of Chikungunya Virus in Bahia, Brazil, 2014-2015. Version 1. *PLoS Curr.* 2016 February 1; 8: PMID: PMC4747681
ecurrents.outbreaks.c97507e3e48efb946401755d468c28b2.

Decision letter (RSPB-2019-0310.R0)

06-Mar-2019

Dear Ms Freitas:

I am writing to inform you that your manuscript RSPB-2019-0310 entitled "Space-time clusters of dengue, chikungunya, and Zika cases in the city of Rio de Janeiro" has, in its current form, been rejected for publication in *Proceedings B*.

This action has been taken on the advice of referees, who have recommended that substantial revisions are necessary. This is not only with regards to the explanations provided of the methods used, but also more fundamentally with regards to clear comments by reviewer 2 and the Associate Editor that they are currently far from convinced that your conclusions are sufficiently robust for publication in *Proceedings B*. With this in mind we would be happy to consider a resubmission, provided the comments of the referees are fully addressed. However please note that this is not a provisional acceptance.

Sincerely,
Proceedings B
mailto: proceedingsb@royalsociety.org

Associate Editor

Comments to Author:

Dear Dr Freitas

your manuscript has now been evaluated by two independent reviewers and myself, and although we all find your study of interest there are some issues about the robustness of your results / conclusions and how much these depend on your assumptions regarding immune competition between the three virus, as opposed to simpler alternatives where viruses are introduced at different time points (see comments by referee #2). If you choose to resubmit a revised manuscript (here or elsewhere) then you may need to better explain and justify your chosen methodology, which will be unfamiliar to many readers, and also pay attention to the comments about data and code accessibility.

With best wishes

Reviewer(s)' Comments to Author:

Referee: 1

Comments to the Author(s)

Freitas et al. present an interesting spatio-temporal analysis of an important triple-epidemic. They provide a generally clear exposition on the use of spatial scan statistics, and raise a number of important points which must be considered for coincident epidemics of these three arboviruses.

We have quite a few, but all relatively minor, comments:

General:

Would you consider making the R code available in an online repository?

Introduction:

Lines 43-44: Whilst the statement regarding the public health & economic burden is probably true, the references given, which focus on individual diseases and are not economic analyses, don't seem relevant to the statement. Are there more relevant ones?

Line 29: Could you consider defining the term 'social vulnerability'?

Methods:

The right panel in figure 1 is hard to read as there are too many things going on in it (its especially hard to tell the population density). Would you be able to make it clearer - perhaps by doing two maps (one for neighborhoods, and one for density)

line 94: could you reference the URL for the data here.

The section about space-time analysis is currently a bit vague - could you please expand a little on the details (perhaps in the supplementary material if there is not room in the main text). In particular:

How did you calculate/define the expected number of cases?

How did you calculate the likelihood ratio for a cluster?

Line 108: should these be the most likely cluster with more observed than expected cases, as in the multivariate situation? (Apologies if this is a misunderstanding!)

line 109-112 is quite confusing. Could you use an equation instead?

Also, why is the risk denominator the expected cases, rather than the population - or is the population proportional to the expected cases in this model?

It looks like the clusters in supp. figure 1 have some overlap, but you state that the clusters should have no spatial overlap in line 121. Please clarify this.

Results:

In the caption for figures 3-5, please explain what the red bands in panel B mean (presumably the time window?). Also, please mention in the caption that clusters are numbered in order of likelihood.

Perhaps consider reporting the p-values of the clusters in tables 1-5

In the supp figure 3, it is not clear which cluster came first. Could you consider labeling the cluster which came first, using a clearer color scale, or using a different color scale for A, B and C? Also, it might help with interpretation to include the region (N, S, W, E) borders on these figures.

Discussion:

Perhaps it would be good to discuss, generally, some of the limitations of spatial scan statistics, to aid readers not familiar with the methodology.

The recent paper by Gordon et al.

(<https://journals.plos.org/plosmedicine/article?id=10.1371/journal.pmed.1002726>, no affiliation with the reviewers) might be relevant to your discussion of dengue-Zika cross-immunity.

The claim in lines 248-250, that herd immunity probably did not have a significant impact on the Zika or dengue dynamics, seems too strong based on the data and references presented. For instance, there is no consideration of the counterfactual situation (of no/lower dengue seroprevalence), or of transmissibility of the viruses. Perhaps remove this claim, or support it. Lines 255-257 please could you provide references for the claims that (a) many researchers questioned if CZS was related to maternal DENV antibodies, and (b) that there is insufficient evidence for this claim.

Line 283: could you provide reference(s) for the claim that the link between poverty and arboviruses is controversial?

Line 300: this statement about misdiagnosis is a little unclear – what exactly are the differences you refer to between? Also, isn't this statement only true if the misdiagnosis is unbiased? For instance, how would misdiagnosis in a particular direction (e.g. Chik often misdiagnosed as Zika, but not vice versa?) affect interpretation? In general, the discussion around misdiagnosis could perhaps be expanded on a little.

Could you perhaps discuss how the detection of joint clusters might differentially affect control, compared to control for single-disease clusters (if you think there would be a difference)?

Is it likely that some of the clusters are spurious? For instance if all of the p-values are close to 0.05 (you have more than 20 clusters).

Referee: 2

Comments to the Author(s)

In this study, the authors explore the geo-temporal patterns of Zika, dengue and chikungunya viruses in Rio de Janeiro, Brazil. The text is well written, with good visualisations of results and description of methods.

My main concerns about this study are (1) the dependence of results interpretation on strong competition between Zika, dengue and chikungunya viruses, (2) as a reader, failing to understand why the method used is ideal to the results presented, and (3) whether the results are innovative or enough for publication in this journal.

(1) I recognize that competition is a topic of debate (whatever the mechanism may be), but as far as I am aware there isn't enough literature to support this as the main driver of the patterns observed. Even if the authors, upon revision, can find enough literature support (which may be possible), the biggest fault at the moment is that alternative / complementary hypothesis / mechanisms are not discussed in the text - for example, different introduction times are the

parsimonious explanation. I recommend the authors discuss these alternatives in their manuscript, as do the authors of the few references included in the current version (see my comments below).

(2) Although the methods are well described, I don't think that the manuscript allows the reader to understand why scan statistics are ideal for this analyses, or why the data requires that method versus a different one (see my comments below).

(3) The results are clear, but the manuscript is essentially a large discussion. My worry is that a small set of results, which overlap with previous literature, may not be enough for this journal. I will nonetheless suggest a major revision and will let the Editor decide on this topic.

"Data" section

Needs a bit more detail. For instance, is geolocation by residence or hospital / medical centre? (later in the discussion is mentioned as residence, but such information should be placed here).

Also, are case counts suspected? confirmed? What are the case definitions in SINAN?

I recognize that the authors have stated the origin of the data, but unless SINAN has very strict publication rules, such data should be made available with this manuscript (for reproducibility purposes, for example).

"Space-time analysis" section

In line 110, what do the authors mean by "expected number of cases within the cluster"?

In line 113, "search" should read "searching".

In line 115, the word 'window' is introduced for the first time - what type of window? time?

The scan statistic approach is based on 'circles' to detect clusters (as stated by the authors). This section should describe how the authors decide if the circle includes a certain region, given that regions are "irregular shapes" (as termed by the authors). How much of an area needs to be included in the circle to be considered part of the cluster? and how is this dealt in light of the fact that different regions have very different areas; and such areas actually seem to have very strong correlation from west to east?

"Results" section

Line 144: why is it interesting that the Zika epidemic does not have a clear peak?

Line 156: "The first dengue cluster in time was detected in the West zone (Supplementary material Figure 3A)." Maybe an misinterpreting the fig S3A, but it seems to show the opposite of what is stated? To show that the first dengue case was in the west, shouldn't there be yellow regions in the West?

At the end of the results I am left without being convinced why the scan statistics method is valuable. That is, it is clear what the method does from the text included, but what this method adds, or what its advantages are in relation to other possible methods is difficult to evaluate. For instance, had the authors simply looked at clusters as groups of adjacent regions (not within circles of a certain radius) with cases in certain time windows, would this have resulted in exactly the same results? This is not a criticism of the method used, but I expect other readers to have the

same doubt. It is important for the authors to discuss this particular topic at some point in the text.

"Discussion" section

line 217: "The number of cases of the three diseases declined after May, coinciding with the end of the rainy and warm season" - this is an opportunity to present such pattern. I suggest the authors include a supplementary figure.

line 220: The authors state: "In a study in Recife, Northeast Brazil, the simultaneous decrease of Zika and increase of chikungunya cases was also observed. The authors interpreted this as a displacement of Zika caused by chikungunya [18]. For Rio de Janeiro city, this might not be the case, as CHIKV caused only a few cases at beginning of 2016, and only started to rise when Zika cases decreased (the depletion of susceptible hosts). Therefore, we hypothesise that ZIKV circulation inhibited CHIKV, rather than CHIKV introduction displacing ZIKV.". The argument of competition between these arboviruses is therefore an essential part of this manuscript. However, I am unsure that this argument is sufficiently supported by the results presented - although competition may be one of the possible solutions. The reference given by the authors ([18]) refers to alternative explanations in Discussion: "(...) These data suggest the displacement of ZIKV by CHIKV in the study area, possibly caused by virus competition in humans and mosquito vectors, and other factors such as acquired immunity to ZIKV in the human population and the high transmission efficiency of CHIKV. Displacement patterns have been observed for distinct DENV serotypes in endemic areas [31, 32] and may occur with distinct arboviruses sharing the same hosts.". It is important to note that the example of DENV in this statement should not apply (in my opinion) to the other arboviruses - it is true that DENV serotypes present displacement patterns, but this is not a consistent observation for the others. Indeed out-of-phase epidemics of these arboviruses is an apparent phenomenon of first epidemics only (see for instance [d]). The authors in [18] further state: "Reported rates of ZIKV/CHIKV co-infections in humans are, in general, low, ranging from 0% to 4.6% [33, 35, 36]. Interestingly, *Ae. aegypti* mosquitoes that are co-infected with ZIKV and CHIKV are capable of transmitting both viruses [37, 38], contradicting in a way the idea of competition. However, viral load of ZIKV in mammalian cells, mosquito cells and whole mosquitoes decrease upon co-infection with ZIKV and CHIKV [37, 38].". Note that alternative and complementary hypothesis / factors are given. In the current manuscript, this is not the case. Finally, it is critical to discuss and consider that the timings, shapes and peaks of these 3 arboviruses are more parsimoniously explained by time of introduction and herd-immunity than they are by competition. If competition was the main biological mechanism dictating the observed patterns, then epidemics in the following years should also present displacement patterns - as mentioned above, this seems not to be the case in most regions. I think this is a critical topic for this manuscript which needs to be better discussed and supported.

line 235: "Not only does *Ae. aegypti* transmit ZIKV at a higher rate, but it is also more easily infected by ZIKV compared to DENV and CHIKV" - this statement should have a citation.

line 240: "Further studies are needed to understand the importance of *Ae. albopictus* in CHIKV transmission." - should this statement include also ZIKV?

line 246: "In our study, the number of dengue cases increased after the peak of Zika cases" - In previous sections the authors refer to the interesting observation that Zika had no peak. This statement should be changed.

line 260: "Dengue, chikungunya, and Zika clusters detected in Rio de Janeiro do not usually coincided in time and space" - this statement needs a citation.

line 263: "In addition to virus interactions and competition for the resources for replication inside the vector, behaviour changes may also impact disease dynamics. A rise in the number of cases may promote vector-control activities, which in turn may decrease the number of cases and hinder the establishment of another arbovirus [25]." - While host behaviour may be one possible driver of this observation, it is again important to offer complementary or alternative hypotheses. For instance, as observed in other studies, is it instead parsimonious that climate dictates or strongly influences the synced end of all 3 epidemics?

line 270: "As dengue has been endemic in Rio de Janeiro for the last three decades and notification of Zika cases was only established in the municipality in October 2015, it was only possible to detect the first disease cluster for chikungunya and pinpoint its source in the North of the city, highlighting once again the importance of interventions in this area." - I find this sentence difficult and am unsure what the intention of the authors is. Please rephrase.

line 273: "The North of Rio de Janeiro has already been identified as a hot spot for dengue and as a key region for dengue diffusion." - this needs a citation.

line 283: "The link between poverty and arbovirus is controversial." - while it may be controversial, many studies have found this relationship with statistical significance. The authors could contribute to make this issue less controversial by citing such studies and spread their findings.

line 293: "Mild cases usually are poorly" should read "are usually"

line 301: "In addition, the extensive experience of health care professionals working in Rio de Janeiro, in detecting and diagnosing dengue symptoms, is thought to reduce the probability of misdiagnosis." - needs citation

Other comments

line 20: "and were not sufficient" should be "have not been sufficient"

lines 22, 23: "Understanding the behaviour of these diseases in a triple epidemic scenario is a necessary step for devising better interventions" - why is it a necessary step?

line 49: "a phenomenon that has been referred to as the 'triple epidemic'" - where has this been shown to be referred this way? please cite.

line 50: "Understanding the behaviour of dengue, Zika, and chikungunya, when they compete in time and space, is a step forward in improving the design of interventions for prevention and outbreak response" - this is one of the statements that assumed the role of competition but that is not really sustained by the text or results in the manuscript.

line 58: "SINAN receives a large number of notifications and it thought to accurately represent the overall trend of the dengue situation in Brazil" - There reference given is in Portuguese and this may be insufficient for a general audience. Plus, it is also the case that other studies have suggested that SINAN (as most passive surveillance systems) does not 'accurately' capture dengue's trends (e.g. [a]). The authors confirm this in the discussion by dwelling around asymptomatic cases and clinical overlap with other viruses.

line 78: What about minimum temperatures? and humidity? these have been shown to be important (e.g. [b,c]) and the authors have an opportunity there to describe the local scenario.

line 244: "in cross-immunity. [23] Whether" - should have [23] before ''

References:

[a] Silva, M. M. O., Rodrigues, M. S., Paploski, I. A. D., Kikuti, M., Kasper, A. M., Cruz, J. S., ... Ribeiro, G. S. (2016). Accuracy of dengue reporting by national surveillance system, Brazil. *Emerging Infectious Diseases*, 22, 336–339.

[b] Lourenço, J., M. Recker, 2014. The 2012 Madeira dengue outbreak: epidemiological determinants and future epidemic potential. *PLoS neglected tropical diseases* 8:e3083

[c] Alto, B. W. and S. A. Juliano, 2001. Precipitation and temperature effects on populations of *Aedes albopictus* (Diptera: Culicidae): Implications for range expansion. *Journal of Medical Entomology* 38:646–656.

[d] Faria et al 2016. Epidemiology of Chikungunya Virus in Bahia, Brazil, 2014–2015. Version 1. *PLoS Curr.* 2016 February 1; 8: PMID: PMC4747681
ecurrents.outbreaks.c97507e3e48efb946401755d468c28b2.

Author's Response to Decision Letter for (RSPB-2019-0310.R0)

See Appendix A.

RSPB-2019-1867.R0

Review form: Reviewer 1

Recommendation

Accept with minor revision (please list in comments)

Scientific importance: Is the manuscript an original and important contribution to its field?

Good

General interest: Is the paper of sufficient general interest?

Good

Quality of the paper: Is the overall quality of the paper suitable?

Good

Is the length of the paper justified?

Yes

Should the paper be seen by a specialist statistical reviewer?

No

Do you have any concerns about statistical analyses in this paper? If so, please specify them explicitly in your report.

No

It is a condition of publication that authors make their supporting data, code and materials available - either as supplementary material or hosted in an external repository. Please rate, if applicable, the supporting data on the following criteria.

Is it accessible?

Yes

Is it clear?

Yes

Is it adequate?

Yes

Do you have any ethical concerns with this paper?

No

Comments to the Author

Thanks for addressing the previous comments. The paper is improved, especially the description of scan statistics and the figures. Thanks also for making your code available and for providing the link to the data.

We have several remaining comments below; line numbers refer to the tracked changes version of the text:

- Main comment: there is still very little discussion of the limitations of scan statistics, although the discussion of strengths is much improved. Please expand on this discussion and describe whether/how such limitations might affect results/interpretation. Relatedly, you might consider moving the limitations of scan statistics (which are currently on lines 351-353, with the rest of the limitations) to the point in the discussion where you discuss the strengths of scan statistics.
 - Lines 107-112: is there a reference for these stated benefits?
 - Line 126: In reference [24], the equation has an indicator function which is 1 if the expected number of cases is greater than the observed number, and zero otherwise, when scanning for high number of cases (as you are in this case, I think). Should there be an indicator function in equation (2)? In the first version of this paper, you stated that, for the multivariate case, the likelihood was 0 if the expected number of cases was below the observed, implying you used the indicator function, but this sentence disappeared in this version. Please clarify whether you are only looking for high numbers of cases (in which case I think there should be an indicator function), or for both high and low numbers.
 - Lines 141-142: what is the difference between 'close together' and 'in similar locations'?
 - Figure 6: could you include the time window as you did for the previous figures?
 - The first paragraph of the discussion currently just mentions why the paper is important; it might benefit from also discussing the key messages of the paper.
 - Line 238: typo: 'singe' should read 'single'.
 - Line 239: you could refer to Magalhaes et al., Insects, 2018 in discussion of sequential infection in mosquitoes and the impact on transmission.
 - Line 282: typo: 'no sufficient' should be 'insufficient'.
 - Line 325: the way the paragraph is structured now, mentioning that the link between poverty and arboviruses is controversial before going on to discuss the evidence for it, is a little

counterintuitive and may be confusing. Perhaps the authors would consider discussing the evidence for the link between arboviruses and poverty first, and perhaps mentioning the controversy or that more evidence is needed at the end of the paragraph? This may better emphasize what evidence there is for this link, make the paragraph less confusing, and help readers to understand your reasoning in the remainder of the discussion, some of which rests on this link.

- Line 326: social vulnerability was removed from the abstract but is still here; perhaps consider rewording as you did in the abstract.
- Were any of the cases co-infected, and would this affect your analysis if they were?
- All of the supplementary figures need captions (there were no captions for the SFs with the proof)
- Supplementary figure 1 should have units for population density.
- SF5 needs labels on the color scale, and titles might be useful on the plots, or labels (A and B) which could be referred to in the caption.

Decision letter (RSPB-2019-1867.R0)

30-Aug-2019

Dear Ms Freitas:

Your manuscript has now been peer reviewed and the reviews have been assessed by an Associate Editor. The reviewers' comments (not including confidential comments to the Editor) and the comments from the Associate Editor are included at the end of this email for your reference. As you will see, the reviewer has raised some concerns with your manuscript and we would like to invite you to revise your manuscript to address them.

Research ethics:

Use of animals and field studies:

Please submit a copy of your revised paper within three weeks. If we do not hear from you within this time your manuscript will be rejected. If you are unable to meet this deadline please let us know as soon as possible, as we may be able to grant a short extension.

Best wishes,

Professor Hans Heesterbeek
 mailto: proceedingsb@royalsociety.org

Associate Editor Board Member

Comments to Author:

Dear Dr Freitas

although the referee seems satisfied with the revision and clarification following the original submission, they still highlighted a number of outstanding issues that will have to be addressed before a final decision can be made - please make sure you respond to each of them.

With best wishes

Reviewer(s)' Comments to Author:

Referee: 1

Comments to the Author(s).

Thanks for addressing the previous comments. The paper is improved, especially the description of scan statistics and the figures. Thanks also for making your code available and for providing the link to the data.

We have several remaining comments below; line numbers refer to the tracked changes version of the text:

- Main comment: there is still very little discussion of the limitations of scan statistics, although the discussion of strengths is much improved. Please expand on this discussion and describe whether/how such limitations might affect results/interpretation. Relatedly, you might consider moving the limitations of scan statistics (which are currently on lines 351-353, with the rest of the limitations) to the point in the discussion where you discuss the strengths of scan statistics.
 - Lines 107-112: is there a reference for these stated benefits?
 - Line 126: In reference [24], the equation has an indicator function which is 1 if the expected number of cases is greater than the observed number, and zero otherwise, when scanning for high number of cases (as you are in this case, I think). Should there be an indicator function in equation (2)? In the first version of this paper, you stated that, for the multivariate case, the likelihood was 0 if the expected number of cases was below the observed, implying you used the indicator function, but this sentence disappeared in this version. Please clarify whether you are only looking for high numbers of cases (in which case I think there should be an indicator function), or for both high and low numbers.
 - Lines 141-142: what is the difference between 'close together' and 'in similar locations'?
 - Figure 6: could you include the time window as you did for the previous figures?
 - The first paragraph of the discussion currently just mentions why the paper is important; it might benefit from also discussing the key messages of the paper.
 - Line 238: typo: 'singe' should read 'single'.
 - Line 239: you could refer to Magalhaes et al., Insects, 2018 in discussion of sequential infection in mosquitoes and the impact on transmission.
 - Line 282: typo: 'no sufficient' should be 'insufficient'.

- Line 325: the way the paragraph is structured now, mentioning that the link between poverty and arboviruses is controversial before going on to discuss the evidence for it, is a little counterintuitive and may be confusing. Perhaps the authors would consider discussing the evidence for the link between arboviruses and poverty first, and perhaps mentioning the controversy or that more evidence is needed at the end of the paragraph? This may better emphasize what evidence there is for this link, make the paragraph less confusing, and help readers to understand your reasoning in the remainder of the discussion, some of which rests on this link.

- Line 326: social vulnerability was removed from the abstract but is still here; perhaps consider rewording as you did in the abstract.

- Were any of the cases co-infected, and would this affect your analysis if they were?

- All of the supplementary figures need captions (there were no captions for the SFs with the proof)

- Supplementary figure 1 should have units for population density.

- SF5 needs labels on the color scale, and titles might be useful on the plots, or labels (A and B) which could be referred to in the caption.

Author's Response to Decision Letter for (RSPB-2019-1867.R0)

See Appendix B.

Decision letter (RSPB-2019-1867.R1)

13-Sep-2019

Dear Ms Freitas

I am pleased to inform you that your manuscript entitled "Space-time dynamics of a triple epidemic: dengue, chikungunya, and Zika clusters in the city of Rio de Janeiro" has been accepted for publication in Proceedings B.

Open Access

You are invited to opt for Open Access, making your freely available to all as soon as it is ready for publication under a CC BY licence. Our article processing charge for Open Access is £1700. Corresponding authors from member institutions

Paper charges

Sincerely,

Professor Hans Heesterbeek
Editor, Proceedings B
mailto: proceedingsb@royalsociety.org

Associate Editor:
Board Member
Comments to Author:
(There are no comments.)

Appendix A

Space-time dynamics of a triple epidemic: dengue, chikungunya, and Zika clusters in the city of Rio de Janeiro

Laís Picinini Freitas, Oswaldo Gonçalves Cruz, Rachel Lowe, Marília Sá Carvalho

Response to referees

COMMENT	RESPONSE
Associate Editor Dear Dr Freitas your manuscript has now been evaluated by two independent reviewers and myself, and although we all find your study of interest there are some issues about the robustness of your results / conclusions and how much these depend on your assumptions regarding immune competition between the three virus, as opposed to simpler alternatives where viruses are introduced at different time points (see comments by referee #2). If you choose to resubmit a revised manuscript (here or elsewhere) then you may need to better explain and justify your chosen methodology, which will be unfamiliar to many readers, and also pay attention to the comments about data and code accessibility. With best wishes	Dear Editor, Please find enclosed a revised version of the manuscript “Space-time clusters of dengue, chikungunya, and Zika cases in the city of Rio de Janeiro” for consideration in Proceedings of the Royal Society B: Biological Sciences. We have carefully addressed all the valuable comments from the editor and reviewers, as detailed below. Besides, we have done a thorough language review, that is not marked in the text, to preserve legibility. We also excluded and summarised part of the manuscript, moving some tables to the supplementary material, to preserve the limit size of 10 pages. We have included more information regarding the chosen methodology. In summary, scan statistic allows disease dynamics to be examined in continuous time, rather than discrete, preselected time windows. This allows us to consider both spatial and temporal dependency structures and test hypotheses using a robust statistical framework. Other advantages are explored in the manuscript. We have explored additional alternatives to explain the results besides competition between viruses, including reintroduction of the chikungunya virus later in the year in 2016, the rapid spread of Zika cases due to a naïve population, and different introduction times of the viruses across the city. We have also made the R code available in Zenodo (in addition to GitHub), and included a link to where the data can be download. We believe the revised manuscript is now suitable for publication in this important journal and will hope it will be of great interest to its readership.

COMMENT	RESPONSE
	Thank you for your consideration. I look forward to your response, With best wishes, Laís Picinini Freitas
Referee #1	
Freitas et al. present an interesting spatio-temporal analysis of an important triple-epidemic. They provide a generally clear exposition on the use of spatial scan statistics, and raise a number of important points which must be considered for coincident epidemics of these three arboviruses. We have quite a few, but all relatively minor, comments:	Thank you for the careful review and comments.
General:  •Would you consider making the R code available in an online repository? 	We are making the R code available online at GitHub and Zonodo: https://github.com/laispfreitas/satscan_dzc/blob/master/script_satscan_dzc_rio DOI 10.5281/zenodo.3362527.
Introduction:	--
 •Lines 43-44: Whilst the statement regarding the public health & economic burden is probably true, the references given, which focus on individual diseases and are not economic analyses, don't seem relevant to the statement. Are there more relevant ones? 	We included more relevant references:  • Cardona-Ospina JA, Villamil-Gómez WE, Jimenez-Canizales CE, Castañeda-Hernández DM, Rodríguez-Morales AJ. 2015 Estimating the burden of disease and the economic cost attributable to chikungunya, Colombia, 2014. Transactions of The Royal Society of Tropical Medicine and Hygiene 109, 793–802. (doi:10.1093/trstmh/trv094) • Lee BY, Alfaro-Murillo JA, Parpia AS, Asti L, Wedlock PT, Hotez PJ, Galvani AP. 2017 The potential economic burden of Zika in the continental United States. PLOS Neglected Tropical Diseases 11, e0005531. (doi:10.1371/journal.pntd.0005531)
 •Line 29: Could you consider defining the term 'social vulnerability'? 	Due to words limits in the abstract, a definition wouldn't fit, so we changed the wording to lower socioeconomic status.
Methods:	--
 •The right panel in figure 1 is hard to read as there are too many things going on in it (its 	Figure 1 now includes only the neighbourhoods and the regions. Population density and green areas

COMMENT	RESPONSE
especially hard to tell the population density). Would you be able to make it clearer - perhaps by doing two maps (one for neighborhoods, and one for density)	were depicted in different maps, included in the supplementary material figure 1.
•line 94: could you reference the URL for the data here.	We included the reference for the data: Prefeitura do Rio de Janeiro, Secretaria Municipal de Saúde. Arboviroses. See http://www.rio.rj.gov.br/web/sms/exibeConteudo?id=6525201 (accessed on 7 August 2019).
• The section about space-time analysis is currently a bit vague – could you please expand a little on the details (perhaps in the supplementary material if there is not room in the main text). In particular:	We have revised this section following the suggestions below.
○ How did you calculate/define the expected number of cases?	For each cylinder, the expected number of cases ($E[c]$) is equal to the total number of cases in the city (C) divided by the total city population (P), times the population within the cylinder (p): $E[c] = \frac{C}{P} \times p$ This was included in the manuscript.
○ How did you calculate the likelihood ratio for a cluster?	The log likelihood ratio (LLR) is calculated for each cluster using the following equation: $LLR = \left(\frac{c}{E[c]}\right)^c \left(\frac{C-c}{C-E[c]}\right)^{C-c}$ where c is the number of cases inside the cluster, and the other letters follow the same notation than the equation for $E[c]$. This has been included in the manuscript.
○ Line 108: should these be the most likely cluster with more observed than expected cases, as in the multivariate situation? (Apologies if this is a misunderstanding!)	The most likely cluster for a single disease is the cluster with the maximum likelihood ratio. For the multivariate scan statistic, the most likely cluster is determined by the maximum of the summed log likelihood ratio for each disease.
○ line 109-112 is quite confusing. Could you use an equation instead?	OK. Included: “The relative risk (RR) for each cluster is calculated using the following equation: $RR = \frac{c/E[c]}{(C-c)/(C-E[c])}$”
○ Also, why is the risk denominator the expected cases, rather than the population – or is the population proportional to the expected cases in this model?	The expected cases are proportional to the population: “For each cylinder, its expected number of cases ($E[c]$) is equal to the total number of cases in the city (C) divided by the total city population (P),

COMMENT	RESPONSE																																								
	times the population within the cylinder (p): $E[c] = \frac{C}{P} \times p$																																								
 It looks like the clusters in supp. figure 1 have some overlap, but you state that the clusters should have no spatial overlap in line 121. Please clarify this. 	SaTScan identifies clusters through moving cylinders. In our analysis, the neighbourhood was considered as part of the cylinder if its centroid was located within the base of the cylinder. We preferred to use the neighbourhoods' borders in Figures 3-6 in the manuscript, but for supplementary material figure 1 we kept the clusters as they are in the standard output of SaTScan. For version 2, we included an explanation in the Methods section: "In our analysis, the neighbourhood was considered as part of the cylinder if its centroid was located within the base of the cylinder." And we included a footnote in the supplementary material figure: "This is the standard output of results from SaTScan. A neighbourhood was considered part of a cluster if its centroid was inside the base of the cylinder (the circle, in this figure)."																																								
Results:	--																																								
 In the caption for figures 3-5, please explain what the red bands in panel B mean (presumably the time window?). Also, please mention in the caption that clusters are numbered in order of likelihood. 	The red band is the time period in which the cluster was detected. We have updated the figure captions accordingly, and indicate that the clusters are numbered in order of likelihood.																																								
 Perhaps consider reporting the p-values of the clusters in tables 1-5 	We included the p-values in a table below, but we have decided not to include this table in the manuscript, as all clusters detected were statistically significant at the 0.05 level (i.e. had had p-value < 0.05).    Cluster Dengue Chikungunya Zika Multivariate     1 1 x 10⁻¹⁷ 1 x 10⁻¹⁷ 1 x 10⁻¹⁷ 1 x 10⁻¹⁷   2 1 x 10⁻¹⁷ 1 x 10⁻¹⁷ 1 x 10⁻¹⁷ 1 x 10⁻¹⁷   3 1 x 10⁻¹⁷ 1 x 10⁻¹⁷ 1 x 10⁻¹⁷ 1 x 10⁻¹⁷   4 1 x 10⁻¹⁷ 1 x 10⁻¹⁷ 1 x 10⁻¹⁷ 1 x 10⁻¹⁷   5 1 x 10⁻¹⁷ 1 x 10⁻¹⁷ 1 x 10⁻¹⁷ 1 x 10⁻¹⁷   6 1 x 10⁻¹⁷ 1 x 10⁻¹⁷ 1 x 10⁻¹⁷ 1 x 10⁻¹⁷   7 1 x 10⁻¹⁷ 1 x 10⁻¹⁷ 1 x 10⁻¹⁷ 1 x 10⁻¹⁷   	Cluster	Dengue	Chikungunya	Zika	Multivariate	1	1 x 10 ⁻¹⁷	1 x 10 ⁻¹⁷	1 x 10 ⁻¹⁷	1 x 10 ⁻¹⁷	2	1 x 10 ⁻¹⁷	1 x 10 ⁻¹⁷	1 x 10 ⁻¹⁷	1 x 10 ⁻¹⁷	3	1 x 10 ⁻¹⁷	1 x 10 ⁻¹⁷	1 x 10 ⁻¹⁷	1 x 10 ⁻¹⁷	4	1 x 10 ⁻¹⁷	1 x 10 ⁻¹⁷	1 x 10 ⁻¹⁷	1 x 10 ⁻¹⁷	5	1 x 10 ⁻¹⁷	1 x 10 ⁻¹⁷	1 x 10 ⁻¹⁷	1 x 10 ⁻¹⁷	6	1 x 10 ⁻¹⁷	1 x 10 ⁻¹⁷	1 x 10 ⁻¹⁷	1 x 10 ⁻¹⁷	7	1 x 10 ⁻¹⁷	1 x 10 ⁻¹⁷	1 x 10 ⁻¹⁷	1 x 10 ⁻¹⁷
Cluster	Dengue	Chikungunya	Zika	Multivariate																																					
1	1 x 10 ⁻¹⁷	1 x 10 ⁻¹⁷	1 x 10 ⁻¹⁷	1 x 10 ⁻¹⁷																																					
2	1 x 10 ⁻¹⁷	1 x 10 ⁻¹⁷	1 x 10 ⁻¹⁷	1 x 10 ⁻¹⁷																																					
3	1 x 10 ⁻¹⁷	1 x 10 ⁻¹⁷	1 x 10 ⁻¹⁷	1 x 10 ⁻¹⁷																																					
4	1 x 10 ⁻¹⁷	1 x 10 ⁻¹⁷	1 x 10 ⁻¹⁷	1 x 10 ⁻¹⁷																																					
5	1 x 10 ⁻¹⁷	1 x 10 ⁻¹⁷	1 x 10 ⁻¹⁷	1 x 10 ⁻¹⁷																																					
6	1 x 10 ⁻¹⁷	1 x 10 ⁻¹⁷	1 x 10 ⁻¹⁷	1 x 10 ⁻¹⁷																																					
7	1 x 10 ⁻¹⁷	1 x 10 ⁻¹⁷	1 x 10 ⁻¹⁷	1 x 10 ⁻¹⁷																																					

COMMENT	RESPONSE																																																							
	 81×10^{-17}1×10^{-17}1×10^{-17}1×10^{-17} 91×10^{-17}1×10^{-17}1×10^{-17}1×10^{-17} 101×10^{-17}1×10^{-17}1×10^{-17}1×10^{-17} 111×10^{-17}1×10^{-17}1×10^{-17}1×10^{-17} 121×10^{-17}1×10^{-17}1×10^{-17}1×10^{-17} 131×10^{-17}1×10^{-17}1×10^{-17}1×10^{-17} 141×10^{-17}7×10^{-11}1×10^{-17}1×10^{-17} 151×10^{-17}0.00021×10^{-17}1×10^{-17} 161×10^{-17}--1×10^{-17} 171.6×10^{-9}--- 180.0032--- 	8	1×10^{-17}	1×10^{-17}	1×10^{-17}	1×10^{-17}	9	1×10^{-17}	1×10^{-17}	1×10^{-17}	1×10^{-17}	10	1×10^{-17}	1×10^{-17}	1×10^{-17}	1×10^{-17}	11	1×10^{-17}	1×10^{-17}	1×10^{-17}	1×10^{-17}	12	1×10^{-17}	1×10^{-17}	1×10^{-17}	1×10^{-17}	13	1×10^{-17}	1×10^{-17}	1×10^{-17}	1×10^{-17}	14	1×10^{-17}	7×10^{-11}	1×10^{-17}	1×10^{-17}	15	1×10^{-17}	0.0002	1×10^{-17}	1×10^{-17}	16	1×10^{-17}	-	-	1×10^{-17}	17	1.6×10^{-9}	-	-	-	18	0.0032	-	-	-
8	1×10^{-17}	1×10^{-17}	1×10^{-17}	1×10^{-17}																																																				
9	1×10^{-17}	1×10^{-17}	1×10^{-17}	1×10^{-17}																																																				
10	1×10^{-17}	1×10^{-17}	1×10^{-17}	1×10^{-17}																																																				
11	1×10^{-17}	1×10^{-17}	1×10^{-17}	1×10^{-17}																																																				
12	1×10^{-17}	1×10^{-17}	1×10^{-17}	1×10^{-17}																																																				
13	1×10^{-17}	1×10^{-17}	1×10^{-17}	1×10^{-17}																																																				
14	1×10^{-17}	7×10^{-11}	1×10^{-17}	1×10^{-17}																																																				
15	1×10^{-17}	0.0002	1×10^{-17}	1×10^{-17}																																																				
16	1×10^{-17}	-	-	1×10^{-17}																																																				
17	1.6×10^{-9}	-	-	-																																																				
18	0.0032	-	-	-																																																				
 In the supp figure 3, it is not clear which cluster came first. Could you consider labeling the cluster which came first, using a clearer color scale, or using a different color scale for A, B and C? Also, it might help with interpretation to include the region (N, S, W, E) borders on these figures. 	Because we want to compare one disease to another, we need to use the same colour scale for A, B, and C. We included red circles to indicate which cluster came first. The region borders were included.																																																							
Discussion:	--																																																							
 Perhaps it would be good to discuss, generally, some of the limitations of spatial scan statistics, to aid readers not familiar with the methodology. 	We included more information about the strengths and limitations of scan statistics in the methods and in the discussion. Methods: “To detect spatio-temporal clusters of arboviral diseases in Rio de Janeiro we used the Kulldorff’s scan statistic. This methodology was chosen as it 1) allows detection of space-time clusters for discrete Poisson probability distributions; 2) tests the statistical significance and corrects for multiple testing; 3) examines disease dynamics in continuous time; 4) estimates the relative risk for each cluster (considering the population); and 5) it can simultaneously evaluate more than one disease.” Discussion: “Scan analysis successfully identified clusters of dengue, chikungunya, and Zika. This method has been used to identify risk areas for arboviral diseases in other locations [36–39]. One of the advantages of this method over commonly used exploratory methods is that it looks for clusters in time continuously, accounting for temporal																																																							

COMMENT	RESPONSE
	dependency, instead of fixed and arbitrary time windows. It also tests for statistical significance, corrects for multiple testing, and estimates the relative risk. The visual and exploratory analysis depends on subjective evaluation, whereas scan statistic methodology is more statistically robust. SaTScan™ is a free and user-friendly tool, which could serve as a valuable disease surveillance tool, particularly in resource-limited settings [40,41].”
 • The recent paper by Gordon et al. (https://journals.plos.org/plosmedicine/article?id=10.1371/journal.pmed.1002726, no affiliation with the reviewers) might be relevant to your discussion of dengue-Zika cross-immunity. 	We have included a statement about the suggested paper in the discussion: “Two recent papers showing results from cohort studies shed some light upon this matter. In a paediatric cohort in Nicaragua, prior DENV infection was associated with lower rates of symptomatic Zika [46], and in a cohort in Pau da Lima, Northeast Brazil, the titres of anti-DENV antibodies before the Zika epidemic were inversely associated with the risk of ZIKV infection [47].”
 • The claim in lines 248-250, that herd immunity probably did not have a significant impact on the Zika or dengue dynamics, seems too strong based on the data and references presented. For instance, there is no consideration of the counterfactual situation (of no/lower dengue seroprevalence), or of transmissibility of the viruses. Perhaps remove this claim, or support it. 	We agree with the referee, and it was removed from the text.
 • Lines 255-257 please could you provide references for the claims that (a) many researchers questioned if CZS was related to maternal DENV antibodies, and (b) that there is insufficient evidence for this claim. 	Included:  • Durbin AP. 2016 Dengue Antibody and Zika: Friend or Foe? Trends in Immunology 37, 635–636. (doi:10.1016/j.it.2016.08.006) • Cohen J. 2017 Dengue may bring out the worst in Zika. Science 355, 1362. (doi:doi: 10.1126/science.355.6332.1362)
 • Line 283: could you provide reference(s) for the claim that the link between poverty and arboviruses is controversial? 	Included:  • Mulligan K, Dixon J, Joanna Sinn C-L, Elliott SJ. 2015 Is dengue a disease of poverty? A systematic review. Pathogens and Global Health 109, 10–18. (doi:10.1179/2047773214Y.0000000168)
 • Line 300: this statement about misdiagnosis is a little unclear – what exactly are the differences you refer to between? Also, isn’t 	We did not find reference discussing the misdiagnosis among the three diseases in Rio de Janeiro. The differential diagnosis among similar

COMMENT	RESPONSE
this statement only true if the misdiagnosis is unbiased? For instance, how would misdiagnosis in a particular direction (e.g. Chik often misdiagnosed as Zika, but not vice versa?) affect interpretation? In general, the discussion around misdiagnosis could perhaps be expanded on a little.	diseases depends on the clinical experience of the medical staff. We decided to exclude part of the text about the clinical experience in Rio de Janeiro since we did not have references for it.
 • Could you perhaps discuss how the detection of joint clusters might differentially affect control, compared to control for single-disease clusters (if you think there would be a difference)? 	We included the type of interventions necessary in such scenario in the discussion: “...the timely allocation of resources to local health services, which can become overloaded, and training of medical teams on the differential diagnosis between the diseases. The preparedness of the health service is also important considering co-infections are possible and clinical outcomes are not clear for such cases [11,52]. Simultaneous clusters also suggest increased exposure to Ae. aegypti and, therefore, vector control activities should also be intensified in these locations.”
 • Is it likely that some of the clusters are spurious? For instance if all of the p-values are close to 0.05 (you have more than 20 clusters) 	No cluster had p-values close to 0.05 (see the table included in this document above).
Referee #2	
In this study, the authors explore the geo-temporal patterns of Zika, dengue and chikungunya viruses in Rio de Janeiro, Brazil. The text is well written, with good visualisations of results and description of methods. My main concerns about this study are (1) the dependence of results interpretation on strong competition between Zika, dengue and chikungunya viruses, (2) as a reader, failing to understand why the method used is ideal to the results presented, and (3) whether the results are innovative or enough for publication in this journal.	Thank you for your careful review and comments.
(1) I recognize that competition is a topic of debate (whatever the mechanism may be), but as far as I am aware there isn't enough literature to support this as the main driver of the patterns observed. Even if the authors, upon revision, can find enough literature support (which may be possible), the biggest fault at the moment is that alternative /	We have now explored different:  • Zika cases rapidly spread due to a naïve population and later, cases decreased due to the depletion of ZIKV susceptible hosts; • The possibility that the prevalence of CHIKV was too low to trigger an epidemic in the beginning of 2016, and that the virus

COMMENT	RESPONSE
complementary hypothesis / mechanisms are not discussed in the text - for example, different introduction times are the parsimonious explanation. I recommend the authors discuss these alternatives in their manuscript, as do the authors of the few references included in the current version (see my comments below).	was subsequently reintroduced to the city;  • Behaviour changes among the human population could be promoting vector-control activities after the number of cases of one arboviral disease increases; • Spatial social inequality: wealthier areas may have better vector-control interventions; • Different introduction times of the viruses across the city.
(2) Although the methods are well described, I don't think that the manuscript allows the reader to understand why scan statistics are ideal for this analyses, or why the data requires that method versus a different one (see my comments below).	We included more information about the strengths and limitations of scan statistics in the methods and in the discussion. Methods: “To detect spatio-temporal clusters of arboviral diseases in Rio de Janeiro we used the Kulldorff’s scan statistic. This methodology was chosen as it 1) allows detection of space-time clusters for discrete Poisson probability distributions; 2) tests the statistical significance and corrects for multiple testing; 3) examines disease dynamics in continuous time; 4) estimates the relative risk for each cluster (considering the population); and 5) it can simultaneously evaluate more than one disease.” Discussion: “Scan analysis successfully identified clusters of dengue, chikungunya, and Zika. This method has been used to identify risk areas for arboviral diseases in other locations [36–39]. One of the advantages of this method over commonly used exploratory methods is that it looks for clusters in time continuously, accounting for temporal dependency, instead of fixed and arbitrary time windows. It also tests for statistical significance, corrects for multiple testing, and estimates the relative risk. The visual and exploratory analysis depends on subjective evaluation, whereas scan statistic methodology is more statistically robust. SaTScan™ is a free and user-friendly tool, which could serve as a valuable disease surveillance tool, particularly in resource-limited settings [40,41].”
(3) The results are clear, but the manuscript is essentially a large discussion. My worry is that a small set of results, which overlap with previous literature, may not be enough for this	We are grateful to the Editor for giving us the opportunity to resubmit this study. To our knowledge, this is the first spatio-temporal analysis of the first Zika and chikungunya epidemics

COMMENT	RESPONSE
journal. I will nonetheless suggest a major revision and will let the Editor decide on this topic.	coinciding with a dengue epidemic, in an intra-urban area. As such, our results do not overlap with previous literature. We present unprecedented results, documenting the spatio-temporal dynamics of a triple epidemic of 76030 cases in a city of 6.3 million inhabitants. We believe the revised manuscript is now suitable for publication in this important journal and will be of great interest to its readership.
### “Data” section	--
Needs a bit more detail. For instance, is geolocation by residence or hospital / medical centre? (later in the discussion is mentioned as residence, but such information should be placed here).	The information has now been included in the first paragraph: “The Municipal Secretariat of Health georeferenced 91% of dengue cases, 95% of chikungunya cases and 92% of Zika cases, using the address of the patient’s residence.”
Also, are case counts suspected? confirmed? What are the case definitions in SINAN?	We analysed notified cases, which are either laboratory or clinically and epidemiologically confirmed. This was included in the text: “We analysed notified cases of dengue, Zika and chikungunya (confirmed by laboratory or by clinical-epidemiological criteria) occurring in Rio de Janeiro municipality between 02 August 2015 and 31 December 2016 (epidemiological weeks 31-2015 and 52-2016), grouped by epidemiological week and neighbourhood of residence. Case definitions follow Ministry of Health protocols [13,20,21].” We also include references for case definitions. The references are in Portuguese and not available in English, as they are guidelines from the Brazilian Ministry of Health.
I recognize that the authors have stated the origin of the data, but unless SINAN has very strict publication rules, such data should be made available with this manuscript (for reproducibility purposes, for example).	We included the reference where the data can be downloaded in the text under Methods→ Data, but we do not have permission to distribute the data. Prefeitura do Rio de Janeiro, Secretaria Municipal de Saúde. Arboviroses. See http://www.rio.rj.gov.br/web/sms/exibeConteudo?id=6525201 (accessed on 7 August 2019).
### “Space-time analysis” section	--
In line 110, what do the authors mean by “expected number of cases within the	The expected number of cases ($E[c]$) is equal to the total number of cases in the city (C) divided by the

COMMENT	RESPONSE
cluster”?	total city population (P), times the population within the cylinder (p): $E[c] = \frac{C}{P} \times p$ This has now been included in the manuscript.
In line 113, “search” should read “searching”.	OK.
In line 115, the word ‘window’ is introduced for the first time - what type of window? time?	The word was changed to ‘cylinder’ for consistency, or to “cluster”, where it was applicable.
The scan statistic approach is based on ‘circles’ to detect clusters (as stated by the authors). This section should describe how the authors decide if the circle includes a certain region, given that regions are “irregular shapes” (as termed by the authors). How much of an area needs to be included in the circle to be considered part of the cluster? and how is this dealt in light of the fact that different regions have very different areas; and such areas actually seem to have very strong correlation from west to east?	The neighbourhood belongs to a ‘circle’ if its centroid lies within the circle. This has been included in the text: “In our analysis, the neighbourhood was considered as part of the cylinder if its centroid was located within the base of the cylinder.” Clusters were limited in terms of size of population, rather than area: “In the output parameters, clusters were restricted to include a maximum of 5% of the population of the city (nearly 315 thousand people).”
### “Results” section	--
Line 144: why is it interesting that the Zika epidemic does not have a clear peak?	Because the epidemic curve did not follow the typical shape for an emerging disease. We removed this comment from the manuscript.
Line 156: “The first dengue cluster in time was detected in the West zone (Supplementary material Figure 3A).” Maybe an misinterpreting the fig S3A, but it seems to show the opposite of what is stated? To show that the first dengue case was in the west, shouldn’t there be yellow regions in the West?	Thank you for this observation. In fact, the first cluster included neighbourhoods from the South and from the West zones. This has been corrected in the text: “The first dengue cluster in time included neighbourhoods located between the South and the West regions” We have included the regions boundaries on the maps to help visualise the composition of each cluster.
At the end of the results I am left without being convinced why the scan statistics method is valuable. That is, it is clear what the method does from the text included, but what this method adds, or what its advantages are in relation to other possible methods is difficult to evaluate. For instance, had the	The advantage of scan statistic is the ability to examine disease dynamics in continuous time, rather than discrete, preselected time windows. This allows us to consider both spatial and temporal dependency structures and test hypotheses, correcting for multiple testing, using a robust statistical framework. Scan statistics also

COMMENT	RESPONSE
authors simply looked at clusters as groups of adjacent regions (not within circles of a certain radius) with cases in certain time windows, would this have resulted in exactly the same results? This is not a criticism of the method used, but I expect other readers to have the same doubt. It is important for the authors to discuss this particular topic at some point in the text.	considers the population, not only the number of cases, and estimates the relative risk. In conclusion, the visual and exploratory analysis depends upon a subjective evaluation, whereas scan statistic is more statistically robust. We included more information about the strengths and limitations of scan statistics in the methods and in the discussion. Methods: “To detect spatio-temporal clusters of arboviral diseases in Rio de Janeiro we used the Kulldorff’s scan statistic. This methodology was chosen as it 1) allows detection of space-time clusters for discrete Poisson probability distributions; 2) tests the statistical significance and corrects for multiple testing; 3) examines disease dynamics in continuous time; 4) estimates the relative risk for each cluster (considering the underlying population); and 5) it can simultaneously evaluate more than one disease.” Discussion: “Scan analysis successfully identified clusters of dengue, chikungunya, and Zika. This method has been used to identify risk areas for arboviral diseases in other locations [36–39]. One of the advantages of this method over commonly used exploratory methods is that it looks for clusters in time continuously, accounting for temporal dependency, instead of fixed and arbitrary time windows. It also tests for statistical significance, corrects for multiple testing, and estimates the relative risk. The visual and exploratory analysis depends on subjective evaluation, whereas scan statistic methodology is more statistically robust. SaTScan™ is a free and user-friendly tool, which could serve as a valuable disease surveillance tool, particularly in resource-limited settings [40,41].”
#### “Discussion” section	--
line 217: “The number of cases of the three diseases declined after May, coinciding with the end of the rainy and warm season” - this is an opportunity to present such pattern. I suggest the authors include a supplementary figure.	We included a figure (supplementary material figure 5) presenting the seasonal and interannual variability patterns of temperature and precipitation in Rio de Janeiro city.

COMMENT

line 220: The authors state: “In a study in Recife, Northeast Brazil, the simultaneous decrease of Zika and increase of chikungunya cases was also observed. The authors interpreted this as a displacement of Zika caused by chikungunya [18]. For Rio de Janeiro city, this might not be the case, as CHIKV caused only a few cases at beginning of 2016, and only started to rise when Zika cases decreased (the depletion of susceptible hosts). Therefore, we hypothesise that ZIKV circulation inhibited CHIKV, rather than CHIKV introduction displacing ZIKV.”. The argument of competition between these arboviruses is therefore an essential part of this manuscript. However, I am unsure that this argument is sufficiently supported by the results presented - although competition may be one of the possible solutions. The reference given by the authors ([18]) refers to alternative explanations in Discussion: “(...) These data suggest the displacement of ZIKV by CHIKV in the study area, possibly caused by virus competition in humans and mosquito vectors, and other factors such as acquired immunity to ZIKV in the human population and the high transmission efficiency of CHIKV. Displacement patterns have been observed for distinct DENV serotypes in endemic areas [31, 32] and may occur with distinct arboviruses sharing the same hosts.”. It is important to note that the example of DENV in this statement should not apply (in my opinion) to the other arboviruses - it is true that DENV serotypes present displacement patterns, but this is not a consistent observation for the others. Indeed out-of-phase epidemics of these arboviruses is an apparent phenomenon of first epidemics only (see for instance [d]). The authors in [18] further state: “Reported rates of ZIKV/CHIKV co-infections in humans are, in general, low, ranging from 0% to 4.6% [33, 35, 36]. Interestingly, *Ae. aegypti* mosquitoes that are co-infected with ZIKV and CHIKV are capable of transmitting both viruses [37, 38], contradicting in a way the idea of competition. However, viral load of ZIKV in mammalian cells, mosquito cells and whole

RESPONSE

In the text highlighted by the referee we had included the depletion of susceptible hosts to ZIKV as an explanation to the decrease of Zika cases, which is equivalent to the acquired immunity factor that the referee cited as present in Magalhães et al. 2017 ([18], in version 1 of our manuscript).

To support their argument that CHIKV displaced ZIKV, Magalhães et al. mention the “the high transmission efficiency of CHIKV”. There is no reference in their paper for this statement, so we could not check. It is our knowledge that *Ae. aegypti* is more successfully infected by ZIKV than CHIKV, and transmits ZIKV at higher rates than CHIKV. This can be checked in Goertz et al. 2017, a study we cited and so did Magalhães et al. (the ref [37] in the text highlighted by the referee). We dedicated a paragraph to this topic in the discussion section. We also included the alternative hypothesis of a subsequent reintroduction of CHIKV in the city:

“The simultaneous decrease of Zika and increase of chikungunya cases was also observed in a study in Recife, Northeast Brazil, and in a study analysing laboratory-confirmed cases in the state of Rio de Janeiro [33,34]. The authors from both studies interpreted this as a displacement of Zika caused by chikungunya. In Rio de Janeiro city, CHIKV was already circulating at the beginning of 2016 but did not trigger an epidemic before Zika cases started decreasing (which was possibly caused by the depletion of ZIKV susceptible hosts). We hypothesise that ZIKV circulation could be inhibiting CHIKV, rather than CHIKV introduction displacing ZIKV. When simultaneously co-infected with both viruses, *Ae. aegypti* was found to transmit ZIKV at a higher rate than CHIKV [35]. The transmission rates for simultaneous co-infection were not significantly different from the rates for single-infection. However, it is not clear how the viruses interact when the mosquito is infected sequentially, not simultaneously. That is, when the mosquito is infected by one virus after biting one person and later by another virus by biting another person, the most likely scenario in the nature considering co-infections in humans are not common [11]. It is also possible that at the beginning of 2016 the

COMMENT	RESPONSE
mosquitoes decrease upon co-infection with ZIKV and CHIKV [37, 38]. ”. Note that alternative and complementary hypothesis / factors are given. In the current manuscript, this is not the case. Finally, it is critical to discuss and consider that the timings, shapes and peaks of these 3 arboviruses are more parsimoniously explained by time of introduction and herd-immunity than they are by competition. If competition was the main biological mechanism dictating the observed patterns, then epidemics in the following years should also present displacement patterns - as mentioned above, this seems not to be the case in most regions. I think this is a critical topic for this manuscript which needs to be better discussed and supported.	prevalence of CHIKV was too low to trigger an epidemic, and that the virus was subsequently reintroduced to the city.” Also regarding alternative hypotheses, in line 262 of version 1 of our manuscript, we already had included: “In addition to virus interactions and competition for the resources for replication inside the vector, behaviour changes may also impact disease dynamics. A rise in the number of cases may promote vector-control activities, which in turn may decrease the number of cases and hinder the establishment of another arbovirus [25]. Also, wealthier areas may have better vector-control interventions, resulting in different spatial distributions.” Because ZIKV and CHIKV started causing epidemics recently, most articles that analysed temporal distributions of the diseases included only the first epidemic waves. Therefore, we believe there is not sufficient evidence on the temporal patterns for the subsequent epidemic waves. For version 2, we included the proposed reference ([d]) and other alternative hypotheses in the discussion. “Alternatively, the observed differences may be a result of different introduction times of the viruses across the city. In a previous study conducted in a large city of Bahia state, Brazil, the chikungunya epidemic curve showed a temporal displacement only in the first wave, synchronizing with the dengue curve in the second wave [51]. We only analysed the first epidemic waves for chikungunya and Zika. Further investigations including subsequent years are important to elucidate if the spatio-temporal distribution of the three diseases changes after epidemic establishment.”
line 235: “Not only does Ae. aegypti transmit ZIKV at a higher rate, but it is also more easily infected by ZIKV compared to DENV and CHIKV” - this statement should have a citation.	We included the citation:  • Göertz GP, Vogels CBF, Geertsema C, Koenraadt CJM, Pijlman GP. 2017 Mosquito co-infection with Zika and chikungunya virus allows simultaneous transmission without affecting vector competence of Aedes aegypti. PLOS

COMMENT	RESPONSE
	Neglected Tropical Diseases 11, e0005654. (doi:10.1371/journal.pntd.0005654)
line 240: “Further studies are needed to understand the importance of Ae. albopictus in CHIKV transmission.” - should this statement include also ZIKV?	We decided to exclude this to focus more on the discussion of our results.
line 246: “In our study, the number of dengue cases increased after the peak of Zika cases” - In previous sections the authors refer to the interesting observation that Zika had no peak. This statement should be changed.	OK. Updated sentence: “In our study, the number of dengue cases increased after the maximum number of Zika cases.”
line 260: “Dengue, chikungunya, and Zika clusters detected in Rio de Janeiro do not usually coincided in time and space” - this statement needs a citation.	This is a result of this manuscript. We corrected the wording. “Dengue, chikungunya, and Zika clusters detected in Rio de Janeiro did not usually coincide in time and space...”
line 263: “In addition to virus interactions and competition for the resources for replication inside the vector, behaviour changes may also impact disease dynamics. A rise in the number of cases may promote vector-control activities, which in turn may decrease the number of cases and hinder the establishment of another arbovirus [25].” - While host behaviour may be one possible driver of this observation, it is again important to offer complementary or alternative hypotheses. For instance, as observed in other studies, is it instead parsimonious that climate dictates or strongly influences the synced end of all 3 epidemics?	We changed the wording in the text to explain that we were talking about human behaviour changes impacting the spatio-temporal distribution of the diseases (not the synced end of all 3 epidemics, which, we agree with the referee and we had previously stated in the text, must be strongly influenced by climate). In fact, the idea was to present behaviour changes among the human population as an alternative/complementary explanation to competition for the observed differences in space-time of the clusters. Updated text: “In addition to virus interactions and competition for the resources for replication inside the vector, behaviour changes among the human population may also help explain the spatio-temporal differences in the distribution of the three diseases. A rise in the number of cases of one arboviral disease may promote vector-control activities, which in turn may decrease the number of cases and hinder the establishment of another arbovirus in that location [50].”
line 270: “As dengue has been endemic in Rio de Janeiro for the last three decades and notification of Zika cases was only established in the municipality in October 2015, it was only possible to detect the first disease cluster for chikungunya and pinpoint its source in the North of the city, highlighting	OK. Updated sentence: “This study captured the first ever reported cases of chikungunya in the city, pinpointing its source in the north of the city. Note, dengue has been

COMMENT	RESPONSE
once again the importance of interventions in this area.” - I find this sentence difficult and am unsure what the intention of the authors is. Please rephrase.	endemic in Rio de Janeiro for the last three decades and notification of Zika cases was only established in the municipality in October 2015 (after the Zika epidemic had already begun).”
line 273: “The North of Rio de Janeiro has already been identified as a hot spot for dengue and as a key region for dengue diffusion.” - this needs a citation.	The citations were in the following sentence. We changed the text to make it clear. The citations:  • Xavier DR, Magalhães M de AFM, Gracie R, Reis IC dos, Matos VP de, Barcellos C. 2017 Difusão espaço-tempo do dengue no Município do Rio de Janeiro, Brasil, no período de 2000-2013. Cadernos de Saúde Pública 33. (doi:10.1590/0102-311x00186615) • Carvalho S, Magalhães MDAFM, Medronho RDA. 2017 Analysis of the spatial distribution of dengue cases in the city of Rio de Janeiro, 2011 and 2012. Revista de Saúde Pública 51. (doi:10.11606/s1518-8787.2017051006239)
line 283: “The link between poverty and arbovirus is controversial.” - while it may be controversial, many studies have found this relationship with statistical significance. The authors could contribute to make this issue less controversial by citing such studies and spread their findings.	In the same paragraph in the version 1 of the manuscript we had already cited the findings of some studies: “In Rio de Janeiro city, areas in or near favelas were detected as hot spots for dengue [30]. Consistent with our findings, a study conducted in French Guiana indicated that, early in the epidemic, the poorest neighbourhoods would have a greater risk for CHIKV infection [32]. In the first dengue epidemic in a city of São Paulo state, Brazil, authors found a direct relationship between low socio-economic conditions and dengue [33].”
line 293: “Mild cases usually are poorly” should read “are usually”	OK.
line 301: “In addition, the extensive experience of health care professionals working in Rio de Janeiro, in detecting and diagnosing dengue symptoms, is thought to reduce the probability of misdiagnosis.” - needs citation	We decided to exclude this part of the text since we did not find references for it.
### Other comments	--
line 20: “and were not sufficient” should be “have not been sufficient”	OK.
lines 22, 23: “Understanding the behaviour of	Current interventions are not effective in the

COMMENT	RESPONSE
these diseases in a triple epidemic scenario is a necessary step for devising better interventions” - why is it a necessary step	control of the incidence of these arboviral diseases. Therefore, it is important to prevent severe clinical developments (such as chronic pain for chikungunya, haemorrhagic dengue fever, and congenital Zika syndrome) and deaths. Given the triple epidemic scenario, the health system needs to be prepared to recognise different symptoms related to each disease and apply clinical interventions suitable for each case, also considering co-infections are possible and clinical outcomes are not clear for such cases. We included this in the introduction and in the discussion. Introduction: “In this scenario, the health care system needs to be prepared to account for medical interventions, different for each disease, and prevent severe clinical developments, also taking into account that co-infections are possible and clinical manifestations for such cases are not clear [11]. Understanding the behaviour of dengue, Zika, and chikungunya, when they co-exist in time and space, is a step forward in improving the design of interventions for prevention and outbreak response [12].” Discussion: “The identification of clusters in space and time allows actions to be intensified in high-risk locations in a timely manner. It is essential that health care facilities are prepared to prevent severe clinical developments (such as haemorrhagic dengue fever, chronic pain among chikungunya cases, and congenital Zika syndrome) and deaths.”
line 49: “a phenomenon that has been referred to as the ‘triple epidemic’” - where has this been shown to be referred this way? please cite.	The term has been used by the Brazilian Ministry of Health, by the media, and by reference researchers for arboviruses, such as Dr. Maria Gloria Teixeira. Most of the texts using the term are in Portuguese, however. We included a citation: Santos DN, Aquino EML, Menezes GM de S, Paim JS, Silva LMV, Souza LEPF, Teixeira MG, Barreto ML. 2016 Documento de posição sobre a tríplice epidemia de Zika-Dengue-Chikungunya.

COMMENT	RESPONSE
line 50: “Understanding the behaviour of dengue, Zika, and chikungunya, when they compete in time and space, is a step forward in improving the design of interventions for prevention and outbreak response” - this is one of the statements that assumed the role of competition but that is not really sustained by the text or results in the manuscript.	We changed the wording to “when they co-exist in time and space”.
line 58: “SINAN receives a large number of notifications and it thought to accurately represent the overall trend of the dengue situation in Brazil” - There reference given is in Portuguese and this may be insufficient for a general audience. Plus, it is also the case that other studies have suggested that SINAN (as most passive surveillance systems) does not ‘accurately’ capture dengue’s trends (e.g. [a]). The authors confirm this in the discussion by dwelling around asymptomatic cases and clinical overlap with other viruses.	We changed the text to make the limitations of SINAN more clear. The reference is in Portuguese, but the abstract is available in English. We also included the reference the referee had indicated. Updated text: “As a passive surveillance system, one of SINAN’s limitation is under-reporting. However, SINAN receives a large number of notifications, and it thought to represent the overall trend of the dengue situation in Brazil [14,15].”
line 78: What about minimum temperatures? and humidity? these have been shown to be important (e.g. [b,c]) and the authors have an opportunity there to describe the local scenario.	We included in the Supplementary material Figure 5 information on minimum temperature and precipitation. We looked for annual average relative humidity for the entire city of Rio de Janeiro, but it was not available.
line 244: “in cross-immunity. [23] Whether” - should have [23] before ‘.’	OK.
### References: [a] Silva, M. M. O., Rodrigues, M. S., Paploski, I. A. D., Kikuti, M., Kasper, A. M., Cruz, J. S., ... Ribeiro, G. S. (2016). Accuracy of dengue reporting by national surveillance system, Brazil. Emerging Infectious Diseases, 22, 336–339. [b] Lourenço, J., M. Recker, 2014. The 2012 Madeira dengue outbreak: epidemiological determinants and future epidemic potential. PLoS neglected tropical diseases 8:e3083 [c] Alto, B. W. and S. A. Juliano, 2001. Precipitation and temperature effects on populations of Aedes albopictus (Diptera: Culicidae): Implications for range expansion. Journal of Medical Entomology 38:646–656.	

COMMENT	RESPONSE
[d] Faria et al 2016. Epidemiology of Chikungunya Virus in Bahia, Brazil, 2014-2015. Version 1. PLoS Curr. 2016 February 1; 8: PMID: PMC4747681 ecurrents.outbreaks.c97507e3e48efb946401755d468c28b2.	

Appendix B

Space-time dynamics of a triple epidemic: dengue, chikungunya, and Zika clusters in the city of Rio de Janeiro

Laís Picinini Freitas, Oswaldo Gonçalves Cruz, Rachel Lowe, Marília Sá Carvalho

Response to referees

COMMENT	RESPONSE
Associate Editor Dear Dr Freitas although the referee seems satisfied with the revision and clarification following the original submission, they still highlighted a number of outstanding issues that will have to be addressed before a final decision can be made - please make sure you respond to each of them. With best wishes	Dear Editor, Please find enclosed a revised version of the manuscript “Space-time dynamics of a triple epidemic: dengue, chikungunya, and Zika clusters in the city of Rio de Janeiro” for consideration in Proceedings of the Royal Society B: Biological Sciences. We have carefully addressed the comments from the reviewer, as detailed below. We believe the revised manuscript is now suitable for publication in this important journal and hope it will be of great interest to its readership. Thank you for your consideration. With best wishes, Laís Picinini Freitas
Referee #1 Thanks for addressing the previous comments. The paper is improved, especially the description of scan statistics and the figures. Thanks also for making your code available and for providing the link to the data. We have several remaining comments below; line numbers refer to the tracked changes version of the text:	Thank you for the careful review and comments.
Main comment: there is still very little discussion of the limitations of scan statistics, although the discussion of strengths is much improved. Please expand on this discussion and describe whether/how such limitations might affect results/interpretation. Relatedly, you might consider moving the limitations of	We moved the limitations of scan statistics as suggested and included more information. “However, the method has some limitations. Scan analysis was not designed to understand diseases trajectory but can be useful in planning interventions. Also, the method detects circular

COMMENT	RESPONSE
scan statistics (which are currently on lines 351-353, with the rest of the limitations) to the point in the discussion where you discuss the strengths of scan statistics.	clusters only, rather than clusters of irregular shapes. Therefore, if a neighbourhood with low risk of the disease is surrounded by neighbourhoods with high risk, it could be considered as part of the cluster. This can be reduced by limiting the size of the clusters.”
Lines 107-112: is there a reference for these stated benefits?	We included a reference. Kulldorff M. 2018 SaTScan™ User Guide for version 9.6.
Line 126: In reference [24], the equation has an indicator function which is 1 if the expected number of cases is greater than the observed number, and zero otherwise, when scanning for high number of cases (as you are in this case, I think). Should there be an indicator function in equation (2)? In the first version of this paper, you stated that, for the multivariate case, the likelihood was 0 if the expected number of cases was below the observed, implying you used the indicator function, but this sentence disappeared in this version. Please clarify whether you are only looking for high numbers of cases (in which case I think there should be an indicator function), or for both high and low numbers.	We changed the text to reflect that we are only interested in clusters with more observed cases than expected. “Through moving cylinders across space (i.e., the base of the cylinder) and time (i.e., the height of the cylinder), it identifies high risk clusters by comparing the observed number of cases to the expected number of cases inside the cylinder [23]”. We also included the indicator function. “The detected clusters are ordered in the results section according to the log likelihood ratio (LLR), such that the cluster with the maximum LLR is the most likely cluster, that is, the cluster least likely to be due to chance. The LLR is calculated using the following equation [24]: $LLR = \left(\frac{c}{E[c]}\right)^c \left(\frac{C-c}{C-E[c]}\right)^{C-c} I() \quad (2)$ where c is the number of cases inside the cluster and I() is an indicator function that is equal to 1 when the cylinder has more cases than expected and 0 otherwise. “
Lines 141-142: what is the difference between ‘close together’ and ‘in similar locations’?	We changed the wording of the sentence: “After testing several combinations of temporal and spatial parameters, we chose the combination that resulted in a reasonable number of clusters that could be targeted for local interventions (Supplementary material Figure 2).”
Figure 6: could you include the time window as you did for the previous figures?	We included the graphs with the time windows as suggested.
The first paragraph of the discussion currently just mentions why the paper is important; it might benefit from also discussing the key messages of the paper.	We included the key messages. “In brief, detected clusters for each disease presented different dynamics in time and space.

COMMENT	RESPONSE
	Dengue and Zika clusters were found across the city, with Zika clusters persisting over a longer time period. Chikungunya clusters were more concentrated in the North and Downtown regions. Simultaneous clusters of the three diseases were more likely in neighbourhoods with a combination of high population density and low socioeconomic status.”
Line 238: typo: ‘singe’ should read ‘single’.	This was corrected in the text.
Line 239: you could refer to Magalhaes et al., Insects, 2018 in discussion of sequential infection in mosquitoes and the impact on transmission.	We included the reference. “However, it is not clear how the viruses interact when the mosquito is infected sequentially, not simultaneously. That is, when the mosquito is infected by one virus after biting one person and later by another virus by biting another person, the most likely scenario in the nature considering co-infections in humans are not common [11]. Under specific laboratory conditions, sequential infection with CHIKV and ZIKV led to enhanced ZIKV transmission [36].”
Line 282: typo: ‘no sufficient’ should be ‘insufficient’.	This was corrected in the text.
Line 325: the way the paragraph is structured now, mentioning that the link between poverty and arboviruses is controversial before going on to discuss the evidence for it, is a little counterintuitive and may be confusing. Perhaps the authors would consider discussing the evidence for the link between arboviruses and poverty first, and perhaps mentioning the controversy or that more evidence is needed at the end of the paragraph? This may better emphasize what evidence there is for this link, make the paragraph less confusing, and help readers to understand your reasoning in the remainder of the discussion, some of which rests on this link.	We moved the sentence about the controversy between arboviral diseases and poverty to the end of the paragraph. “The North of the city is marked by a combination of high population density and a lower HDI than the city average [16]. In Rio de Janeiro, areas in or near favelas were detected as hot spots for dengue [55]. Consistent with our findings, a study conducted in French Guiana indicated that, early in the epidemic, the poorest neighbourhoods would have a greater risk for CHIKV infection [56]. In the first dengue epidemic in a city of São Paulo state, Brazil, authors found a direct relationship between low socio-economic conditions and dengue [57]. We did not observe this relationship for dengue possibly because dengue has already had sustained transmission in the city for decades. The link between poverty and arbovirus is controversial [58]. Nonetheless, locations with social and economic vulnerability more likely have poorer sanitary conditions and less efficient vector-control interventions, which would facilitate mosquito proliferation.”

COMMENT	RESPONSE
Line 326: social vulnerability was removed from the abstract but is still here; perhaps consider rewording as you did in the abstract.	We reworded to “low socioeconomic status”.
Were any of the cases co-infected, and would this affect your analysis if they were?	We do not have information on coinfection cases in the dataset we used. Because coinfections are rare, it should not affect the analysis. We included this in the limitations of the study. “Also, we did not have information on co-infections within the disease surveillance database. However, as co-infections are rare, this should not had affected our analysis. In a national survey in Colombia, co-infections accounted for 0.14% of the arboviral diseases cases [59].”
All of the supplementary figures need captions (there were no captions for the SFs with the proof)	We apologize for that. Titles and captions for supplementary figures had been provided in the respective fields in “Step 5: Details & Comments”. To prevent this from happening again, we are uploading all supplementary figures in a single pdf file with all titles and captions.
Supplementary figure 1 should have units for population density.	We included the units in the supplementary figure 1.
SF5 needs labels on the color scale, and titles might be useful on the plots, or labels (A and B) which could be referred to in the caption.	We included the labels, captions and titles.